# On the Robustness of Safe Reinforcement Learning under Observational Perturbations

**Zuxin Liu[1], Zijian Guo[1], Zhepeng Cen[1], Huan Zhang[1], Jie Tan[2], Bo Li[3], Ding Zhao[1]**
[1]CMU, [2] Google Brain, [3] UIUC
`{zuxinl, zijiang, zcen, huanzhan}@andrew.cmu.edu`
`jietan@google.com, lbo@illinois.edu, dingzhao@cmu.edu`

## Abstract

Safe reinforcement learning (RL) trains a policy to maximize the task reward while satisfying safety constraints. While prior works focus on the performance optimality, we find that the optimal solutions of many safe RL problems are not robust and safe against carefully designed observational perturbations. We formally analyze the unique properties of designing effective observational adversarial attackers in the safe RL setting. We show that baseline adversarial attack techniques for standard RL tasks are not always effective for safe RL and propose two new approaches - one maximizes the cost and the other maximizes the reward. One interesting and counter-intuitive finding is that the maximum reward attack is strong, as it can both induce unsafe behaviors and make the attack stealthy by maintaining the reward. We further propose a robust training framework for safe RL and evaluate it via comprehensive experiments. This paper provides a pioneer work to investigate the safety and robustness of RL under observational attacks for future safe RL studies. Code is available at: `https://github.com/liuzuxin/safe-rl-robustness`

## 1 Introduction

Despite the great success of deep reinforcement learning (RL) in recent years, it is still challenging to ensure safety when deploying them to the real world. Safe RL tackles the problem by solving a constrained optimization that can maximize the task reward while satisfying safety constraints (Brunke et al., 2021), which has shown to be effective in learning a safe policy in many tasks (Zhao et al., 2021; Liu et al., 2022; Sootla et al., 2022b). The success of recent safe RL approaches leverages the power of neural networks (Srinivasan et al., 2020; Thananjeyan et al., 2021). However, it has been shown that neural networks are vulnerable to adversarial attacks – a small perturbation of the input data may lead to a large variance of the output (Machado et al., 2021; Pitropakis et al., 2019), which raises a concern when deploying a neural network RL policy to safety-critical applications (Akhtar & Mian, 2018). While many recent safe RL methods with deep policies can achieve outstanding constraint satisfaction in noise-free simulation environments, such a concern regarding their vulnerability under adversarial perturbations has not been studied in the safe RL setting. We consider the observational perturbations that commonly exist in the physical world, such as unavoidable sensor errors and upstream perception inaccuracy (Zhang et al., 2020a).

Several recent works of observational robust RL have shown that deep RL agent could be attacked via sophisticated observation perturbations, drastically decreasing their rewards (Huang et al., 2017; Zhang et al., 2021). However, the robustness concept and adversarial training methods in standard RL settings may not be suitable for safe RL because of an additional metric that characterizes the cost of constraint violations (Brunke et al., 2021). The cost should be more important than the measure of reward, since any constraint violations could be fatal and unacceptable in the real world (Berkenkamp et al., 2017). For example, consider the autonomous vehicle navigation task where the reward is to reach the goal as fast as possible and the safety constraint is to not collide with obstacles, then sacrificing some reward is not comparable with violating the constraint because the latter may cause catastrophic consequences. However, we find little research formally studying the robustness in the safe RL setting with adversarial observation perturbations, while we believe this should be an important aspect in the safe RL area, because **a vulnerable policy under adversarial attacks cannot be regarded as truly safe in the physical world.**

We aim to address the following questions in this work: 1) How vulnerable would a learned RL agent be under observational adversarial attacks? 2) How to design effective attackers in the safe RL setting? 3) How to obtain a robust policy that can maintain safety even under worst-case perturbations? To answer them, we formally define the observational robust safe RL problem and discuss how to evaluate the adversary and robustness of a safe RL policy. We also propose two strong adversarial attacks that can induce the agent to perform unsafe behaviors and show that adversarial training can help improve the robustness of constraint satisfaction. We summarize the contributions as follows.

1. We formally analyze the policy vulnerability in safe RL under observational corruptions, investigate the observational-adversarial safe RL problem, and show that the optimal solutions of safe RL problems are vulnerable under observational adversarial attacks.

2. We find that existing adversarial attacks focusing on minimizing agent rewards do not always work, and propose two effective attack algorithms with theoretical justifications – one directly maximizes the cost, and one maximizes the task reward to induce a tempting but risky policy. Surprisingly, the maximum reward attack is very strong in inducing unsafe behaviors, both in theory and practice. We believe this property is overlooked as maximizing reward is the optimization goal for standard RL, yet it leads to risky and stealthy attacks to safety constraints.

3. We propose an adversarial training algorithm with the proposed attackers and show contraction properties of their Bellman operators. Extensive experiments in continuous control tasks show that our method is more robust against adversarial perturbations in terms of constraint satisfaction.

## 2 RELATED WORK

**Safe RL.** One type of approach utilizes domain knowledge of the target problem to improve the safety of an RL agent, such as designing a safety filter (Dalal et al., 2018; Yu et al., 2022), assuming sophisticated system dynamics model (Liu et al., 2020; Luo & Ma, 2021; Chen et al., 2021), or incorporating expert interventions (Saunders et al., 2017; Alshiekh et al., 2018). Constrained Markov Decision Process (CMDP) is a commonly used framework to model the safe RL problem, which can be solved via constrained optimization techniques (García & Fernández, 2015; Gu et al., 2022; Sootla et al., 2022a; Flet-Berliac & Basu, 2022). The Lagrangian-based method is a generic constrained optimization algorithm to solve CMDP, which introduces additional Lagrange multipliers to penalize constraints violations (Bhatnagar & Lakshmanan, 2012; Chow et al., 2017; As et al., 2022). The multiplier can be optimized via gradient descent together with the policy parameters (Liang et al., 2018; Tessler et al., 2018), and can be easily incorporated into many existing RL methods (Ray et al., 2019). Another line of work approximates the non-convex constrained optimization problem with low-order Taylor expansions and then obtains the dual variable via convex optimization (Yu et al., 2019; Yang et al., 2020; Gu et al., 2021; Kim & Oh, 2022). Since the constrained optimization-based methods are more general, we will focus on the discussions of safe RL upon them.

**Robust RL.** The robustness definition in the RL context has many interpretations (Sun et al., 2021; Moos et al., 2022; Korkmaz, 2023), including the robustness against action perturbations (Tessler et al., 2019), reward corruptions (Wang et al., 2020; Lin et al., 2020; Eysenbach & Levine, 2021), domain shift (Tobin et al., 2017; Muratore et al., 2018), and dynamics uncertainty (Pinto et al., 2017; Huang et al., 2022). The most related works are investigating the observational robustness of an RL agent under observational adversarial attacks (Zhang et al., 2020a; 2021; Liang et al., 2022; Korkmaz, 2022). It has been shown that the neural network policies can be easily attacked by adversarial observation noise and thus lead to much lower rewards than the optimal policy (Huang et al., 2017; Kos & Song, 2017; Lin et al., 2017; Pattanaik et al., 2017). However, most of the robust RL approaches model the attack and defense regarding the reward, while the robustness regarding safety, i.e., constraint satisfaction for safe RL, has not been formally investigated.

## 3 OBSERVATIONAL ADVERSARIAL ATTACK FOR SAFE RL

### 3.1 MDP, CMDP, AND THE SAFE RL PROBLEM

An infinite horizon Markov Decision Process (MDP) is defined by the tuple $(\mathcal{S}, \mathcal{A}, \mathcal{P}, r, \gamma, \mu_0)$, where $\mathcal{S}$ is the state space, $\mathcal{A}$ is the action space, $\mathcal{P} : \mathcal{S} \times \mathcal{A} \times \mathcal{S} \to [0, 1]$ is the transition kernel that specifies the transition probability $p(s_{t+1}|s_t, a_t)$ from state $s_t$ to $s_{t+1}$ under the action $a_t$, $r : \mathcal{S} \times \mathcal{A} \times \mathcal{S} \to \mathbb{R}$ is the reward function, $\gamma \to [0, 1)$ is the discount factor, and $\mu_0 : \mathcal{S} \to [0, 1]$ is the initial state distribution. We study safe RL under the Constrained MDP (CMDP) framework

$\mathcal{M} \coloneqq (\mathcal{S}, \mathcal{A}, \mathcal{P}, r, c, \gamma, \mu_0)$ with an additional element $c : \mathcal{S} \times \mathcal{A} \times \mathcal{S} \to [0, C_m]$ to characterize the cost for violating the constraint, where $C_m$ is the maximum cost (Altman, 1998).

We denote a safe RL problem as $\mathcal{M}_\Pi^\kappa$, where $\Pi : \mathcal{S} \times \mathcal{A} \to [0, 1]$ is the policy class, and $\kappa \to [0, +\infty)$ is the cost threshold. Let $\pi(a|s) \in \Pi$ denote the policy and $\tau = \{s_0, a_0, ..., \}$ denote the trajectory. We use shorthand $f_t = f(s_t, a_t, s_{t+1}), f \in \{r, c\}$ for simplicity. The value function is $V_f^\pi(\mu_0) = \mathbb{E}_{\tau \sim \pi, s_0 \sim \mu_0}[\sum_{t=0}^\infty \gamma^t f_t]$, which is the expectation of discounted return under the policy $\pi$ and the initial state distribution $\mu_0$. We overload the notation $V_f^\pi(s) = \mathbb{E}_{\tau \sim \pi, s_0 = s}[\sum_{t=0}^\infty \gamma^t f_t]$ to denote the value function with the initial state $s_0 = s$, and denote $Q_f^\pi(s, a) = \mathbb{E}_{\tau \sim \pi, s_0 = s, a_0 = a}[\sum_{t=0}^\infty \gamma^t f_t]$ as the state-action value function under the policy $\pi$. The objective of $\mathcal{M}_\Pi^\kappa$ is to find the policy that maximizes the reward while limiting the cost under threshold $\kappa$:

$$\pi^* = \arg\max_\pi V_r^\pi(\mu_0), \quad s.t. \quad V_c^\pi(\mu_0) \leq \kappa. \tag{1}$$

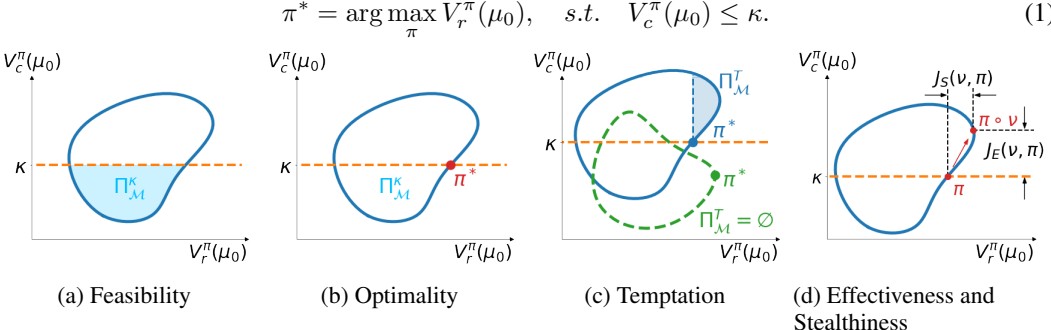

|  |  |  |  |
|---|---|---|---|
| (a) Feasibility | (b) Optimality | (c) Temptation | (d) Effectiveness and Stealthiness |

Figure 1: Illustration of definitions via a mapping from the policy space to the metric plane $\Pi \to \mathbb{R}^2$, where the x-axis is the reward return and the y-axis is the cost return. A point on the metric plane denotes corresponding policies, i.e., the point $(v_r, v_c)$ represents the policies $\{\pi \in \Pi | V_r^\pi(\mu_0) = v_r, V_c^\pi(\mu_0) = v_c\}$. The blue and green circles denote the policy space of two safe RL problems.

We then define feasibility, optimality and temptation to better describe the properties of a safe RL problem $\mathcal{M}_\Pi^\kappa$. The figure illustration of one example is shown in Fig. 1. Note that although the temptation concept naturally exists in many safe RL settings under the CMDP framework, we did not find formal descriptions or definitions of it in the literature.

**Definition 1. Feasibility**. The feasible policy class is the set of policies that satisfies the constraint with threshold $\kappa$: $\Pi_\mathcal{M}^\kappa \coloneqq \{\pi(a|s) : V_c^\pi(\mu_0) \leq \kappa, \pi \in \Pi\}$. A feasible policy should satisfy $\pi \in \Pi_\mathcal{M}^\kappa$.

**Definition 2. Optimality**. A policy $\pi^*$ is optimal in the safe RL context if 1) it is feasible: $\pi^* \in \Pi_\mathcal{M}^\kappa$; 2) no other feasible policy has higher reward return than it: $\forall \pi \in \Pi_\mathcal{M}^\kappa, V_r^{\pi^*}(\mu_0) \geq V_r^\pi(\mu_0)$.

We denote $\pi^*$ as the optimal policy. Note that the optimality is defined w.r.t. the reward return within the feasible policy class $\Pi_\mathcal{M}^\kappa$ rather than the full policy class space $\Pi$, which means that policies that have a higher reward return than $\pi^*$ may exist in a safe RL problem due to the constraint, and we formally define them as tempting policies because they are rewarding but unsafe:

**Definition 3. Temptation**. We define the tempting policy class as the set of policies that have a higher reward return than the optimal policy: $\Pi_\mathcal{M}^T \coloneqq \{\pi(a|s) : V_r^\pi(\mu_0) > V_r^{\pi^*}(\mu_0), \pi \in \Pi\}$. A tempting safe RL problem has a non-empty tempting policy class: $\Pi_\mathcal{M}^T \neq \emptyset$.

We show that all the tempting policies are not feasible (proved by contradiction in Appendix A.1):

**Lemma 1.** *The tempting policy class and the feasible policy class are disjoint: $\Pi_\mathcal{M}^T \cap \Pi_\mathcal{M}^\kappa = \emptyset$. Namely, all the tempting policies violate the constraint: $\forall \pi \in \Pi_\mathcal{M}^T, V_c^\pi(\mu_0) > \kappa$.*

The existence of tempting policies is a unique feature, and one of the major challenges of safe RL since the agent needs to maximize the reward carefully to avoid being tempted. One can always tune the threshold $\kappa$ to change the temptation status of a safe RL problem with the same CMDP. In this paper, we only consider the solvable **tempting** safe RL problems because otherwise, the non-tempting safe RL problem $\mathcal{M}_\Pi^\kappa$ can be reduced to a standard RL problem – an optimal policy could be obtained by maximizing the reward without considering the constraint.

### 3.2 Safe RL under observational perturbations

We introduce a deterministic **observational** adversary $\nu(s) : \mathcal{S} \to \mathcal{S}$ which corrupts the state observation of the agent. We denote the corrupted observation as $\tilde{s} \coloneqq \nu(s)$ and the corrupted policy

as $\pi \circ \nu := \pi(a|\tilde{s}) = \pi(a|\nu(s))$, as the state is first contaminated by $\nu$ and then used by the operator $\pi$. Note that the adversary does **not** modify the original CMDP and true states in the environment, but only the input of the agent. This setting mimics realistic scenarios, for instance, the adversary could be the noise from the sensing system or the errors from the upstream perception system.

Constraint satisfaction is of the top priority in safe RL, since violating constraints in safety-critical applications can be unaffordable. In addition, the reward metric is usually used to measure the agent's performance in finishing a task, so significantly reducing the task reward may warn the agent of the existence of attacks. As a result, a strong adversary in the safe RL setting aims to generate more constraint violations while maintaining high rewards to make the attack stealthy. In contrast, existing adversaries on standard RL aim to reduce the overall reward. Concretely, we evaluate the adversary performance for safe RL from two perspectives:

**Definition 4. (Attack) Effectiveness** $J_E(\nu, \pi)$ is defined as the increased cost value under the adversary: $J_E(\nu, \pi) = V_c^{\pi \circ \nu}(\mu_0) - V_c^{\pi}(\mu_0)$. An adversary $\nu$ is effective if $J_E(\nu, \pi) > 0$.

The effectiveness metric measures an adversary's capability of attacking the safe RL agent to violate constraints. We additionally introduce another metric to characterize the adversary's stealthiness w.r.t. the task reward in the safe RL setting.

**Definition 5. (Reward) Stealthiness** $J_S(\nu, \pi)$ is defined as the increased reward value under the adversary: $J_S(\nu, \pi) = V_r^{\pi \circ \nu}(\mu_0) - V_r^{\pi}(\mu_0)$. An adversary $\nu$ is stealthy if $J_S(\nu, \pi) \geq 0$.

Note that the stealthiness concept is widely used in supervised learning (Sharif et al., 2016; Pitropakis et al., 2019). It usually means that the adversarial attack should be covert to human eyes regarding the input data so that it can hardly be identified (Machado et al., 2021). While the stealthiness regarding the perturbation range is naturally satisfied based on the perturbation set definition, we introduce another level of stealthiness in terms of the task reward in the safe RL task. In some situations, the agent might easily detect a dramatic reward drop. A more stealthy attack is maintaining the agent's task reward while increasing constraint violations; see Appendix B.1 for more discussions.

In practice, the power of the adversary is usually restricted (Madry et al., 2017; Zhang et al., 2020a), such that the perturbed observation will be limited within a pre-defined perturbation set $B(s)$: $\forall s \in \mathcal{S}, \nu(s) \in B(s)$. Following convention, we define the perturbation set $B_p^{\epsilon}(s)$ as the $\ell_p$-ball around the original observation: $\forall s' \in B_p^{\epsilon}(s), \|s' - s\|_p \leq \epsilon$, where $\epsilon$ is the ball size.

### 3.3 VULNERABILITY OF AN OPTIMAL POLICY UNDER ADVERSARIAL ATTACKS

We aim to design strong adversaries such that they are effective in making the agent unsafe and keep reward stealthiness. Motivated by Lemma 1, we propose the **Maximum Reward (MR) attacker** that corrupts the observation by maximizing the reward value: $\nu_{\text{MR}} = \arg\max_{\nu} V_r^{\pi \circ \nu}(\mu_0)$

**Proposition 1.** *For an optimal policy $\pi^* \in \Pi$, the MR attacker is guaranteed to be reward stealthy and effective, given enough large perturbation set $B_p^{\epsilon}(s)$ such that $V_r^{\pi^* \circ \nu_{MR}} > V_r^{\pi^*}$.*

The MR attacker is counter-intuitive because it is exactly the goal for standard RL. This is an interesting phenomenon worthy of highlighting since we observe that the MR attacker effectively makes the optimal policy unsafe and retains stealthy regarding the reward in the safe RL setting. The proof is given in Appendix A.1. If we enlarge the policy space from $\Pi : \mathcal{S} \times \mathcal{A} \to [0, 1]$ to an augmented space $\bar{\Pi} : \mathcal{S} \times \mathcal{A} \times \mathcal{O} \to [0, 1]$, where $\mathcal{O} = \{0, 1\}$ is the space of indicator, we can further observe the following important property for the optimal policy:

**Lemma 2.** *The optimal policy $\pi^* \in \bar{\Pi}$ of a tempting safe RL problem satisfies: $V_c^{\pi^*}(\mu_0) = \kappa$.*

The proof is given in Appendix A.2. The definition of the augmented policy space is commonly used in hierarchical RL and can be viewed as a subset of option-based RL (Riemer et al., 2018; Zhang & Whiteson, 2019). Note that Lemma 2 holds in expectation rather than for a single trajectory. It suggests that the optimal policy in a tempting safe RL problem will be vulnerable as it is on the safety boundary, which motivates us to propose the **Maximum Cost (MC) attacker** that corrupts the observation of a policy $\pi$ by maximizing the cost value: $\nu_{\text{MC}} = \arg\max_{\nu} V_c^{\pi \circ \nu}(\mu_0)$

It is apparent to see that the MC attacker is effective w.r.t. the optimal policy with a large enough perturbation range, since we directly solve the adversarial observation such that it can maximize the constraint violations. Therefore, as long as $\nu_{\text{MC}}$ can lead to a policy that has a higher cost return than $\pi^*$, it is guaranteed to be effective in making the agent violate the constraint based on Lemma 2.

Practically, given a fixed policy $\pi$ and its critics $Q_f^\pi(s, a), f \in \{r, c\}$, we obtain the corrupted observation $\tilde{s}$ of $s$ from the MR and MC attackers by solving:

$$\nu_{\text{MR}}(s) = \arg \max_{\tilde{s} \in B_p^\epsilon(s)} \mathbb{E}_{\tilde{a} \sim \pi(a|\tilde{s})} [Q_r^\pi(s, \tilde{a})], \quad \nu_{\text{MC}}(s) = \arg \max_{\tilde{s} \in B_p^\epsilon(s)} \mathbb{E}_{\tilde{a} \sim \pi(a|\tilde{s})} [Q_c^\pi(s, \tilde{a})] \quad (2)$$

Suppose the policy $\pi$ and the critics $Q$ are all parametrized by differentiable models such as neural networks, then we can back-propagate the gradient through $Q$ and $\pi$ to solve the adversarial observation $\tilde{s}$. This is similar to the policy optimization procedure in DDPG (Lillicrap et al., 2015), whereas we replace the optimization domain from the policy parameter space to the observation space $B_p^\epsilon(s)$. The attacker implementation details can be found in Appendix C.1.

### 3.4 THEORETICAL ANALYSIS OF ADVERSARIAL ATTACKS

**Theorem 1** (Existence of optimal and deterministic MC/MR attackers). *A deterministic MC attacker $\nu_{MC}$ and a deterministic MR attacker $\nu_{MR}$ always exist, and there is no stochastic adversary $\nu'$ such that $V_c^{\pi \circ \nu'}(\mu_0) > V_c^{\pi \circ \nu_{MC}}(\mu_0)$ or $V_r^{\pi \circ \nu'}(\mu_0) > V_r^{\pi \circ \nu_{MR}}(\mu_0)$.*

Theorem 1 provides the theoretical foundation of Bellman operators that require optimal and deterministic adversaries in the next section. The proof is given in Appendix A.3. We can also obtain the upper-bound of constraint violations of the adversary attack at state $s$. Denote $\mathcal{S}_c$ as the set of unsafe states that have non-zero cost: $\mathcal{S}_c := \{s' \in \mathcal{S} : c(s, a, s') > 0\}$ and $p_s$ as the maximum probability of entering unsafe states from state $s$: $p_s = \max_a \sum_{s' \in \mathcal{S}_c} p(s'|s, a)$.

**Theorem 2** (One-step perturbation cost value bound). *Suppose the optimal policy is locally $L$-Lipschitz continuous at state $s$: $D_{\text{TV}}[\pi(\cdot|s')\|\pi(\cdot|s)] \leq L \|s' - s\|_p$, and the perturbation set of the adversary $\nu(s)$ is an $\ell_p$-ball $B_p^\epsilon(s)$. Let $\tilde{V}_c^{\pi,\nu}(s) = \mathbb{E}_{a \sim \pi(\cdot|\nu(s)), s' \sim p(\cdot|s,a)}[c(s, a, s') + \gamma V_c^\pi(s')]$ denote the cost value for only perturbing state $s$. The upper bound of $\tilde{V}_c^{\pi,\nu}(s)$ is given by:*

$$\tilde{V}_c^{\pi,\nu}(s) - V_c^\pi(s) \leq 2L\epsilon \left( p_s C_m + \frac{\gamma C_m}{1 - \gamma} \right). \quad (3)$$

Note that $\tilde{V}_c^{\pi,\nu}(s) \neq V_c^\pi(\nu(s))$ because the next state $s'$ is still transited from the original state $s$, i.e., $s' \sim p(\cdot|s, a)$ instead of $s' \sim p(\cdot|\nu(s), a)$. Theorem 2 indicates that the power of an adversary is controlled by the policy smoothness $L$ and perturbation range $\epsilon$. In addition, the $p_s$ term indicates that a safe policy should keep a safe distance from the unsafe state to prevent it from being attacked. We further derive the upper bound of constraint violation for attacking the entire episodes.

**Theorem 3** (Episodic bound). *Given a feasible policy $\pi \in \Pi_{\mathcal{M}}^\kappa$, suppose $L$-Lipschitz continuity holds globally for $\pi$, and the perturbation set is an $\ell_p$-ball, then the following bound holds:*

$$V_c^{\pi \circ \nu}(\mu_0) \leq \kappa + 2L\epsilon C_m \left( \frac{1}{1 - \gamma} + \frac{4\gamma L\epsilon}{(1 - \gamma)^2} \right) \left( \max_s p_s + \frac{\gamma}{1 - \gamma} \right). \quad (4)$$

See Theorem 2, 3 proofs in Appendix A.4, A.5. We can still observe that the maximum cost value under perturbations is bounded by the Lipschitzness of the policy and the maximum perturbation range $\epsilon$. The bound is tight since when $\epsilon \to 0$ (no attack) or $L \to 0$ (constant policy $\pi(\cdot|s)$ for all states), the RHS is 0 for Eq. (3) and $\kappa$ for Eq. (4), which means that the attack is ineffective.

## 4 OBSERVATIONAL ROBUST SAFE RL

### 4.1 ADVERSARIAL TRAINING AGAINST OBSERVATIONAL PERTURBATIONS

To defend against observational attacks, we propose an adversarial training method for safe RL. We directly optimize the policy upon the corrupted sampling trajectories $\tilde{\tau} = \{s_0, \tilde{a}_0, s_1, \tilde{a}_1, ...\}$, where $\tilde{a}_t \sim \pi(a|\nu(s_t))$. We can compactly represent the adversarial safe RL objective under $\nu$ as:

$$\pi^* = \arg \max_\pi V_r^{\pi \circ \nu}(\mu_0), \quad s.t. \quad V_c^{\pi \circ \nu}(\mu_0) \leq \kappa, \quad \forall \nu. \quad (5)$$

The adversarial training objective (5) can be solved by many policy-based safe RL methods, such as the primal-dual approach, and we show that the Bellman operator for evaluating the policy performance under a deterministic adversary is a contraction (see Appendix A.6 for proof).

**Theorem 4** (Bellman contraction). *Define the Bellman policy operator as $\mathcal{T}_\pi : \mathbb{R}^{|\mathcal{S}|} \to \mathbb{R}^{|\mathcal{S}|}$:*

$$(\mathcal{T}_\pi V_f^{\pi \circ \nu})(s) = \sum_{a \in \mathcal{A}} \pi(a|\nu(s)) \sum_{s' \in \mathcal{S}} p(s'|s,a) \left[ f(s,a,s') + \gamma V_f^{\pi \circ \nu}(s') \right], \quad f \in \{r,c\}. \tag{6}$$

*The Bellman equation can be written as $V_f^{\pi \circ \nu}(s) = (\mathcal{T}_\pi V_f^{\pi \circ \nu})(s)$. In addition, the operator $\mathcal{T}_\pi$ is a contraction under the sup-norm $\|\cdot\|_\infty$ and has a fixed point.*

Theorem 4 shows that we can accurately evaluate the task performance (reward return) and the safety performance (cost return) of a policy under one fixed deterministic adversary, which is similar to solving a standard CMDP. The Bellman contraction property provides the theoretical justification of adversarial training, i.e., training a safe RL agent under observational perturbed sampling trajectories. Then the key part is selecting proper adversaries during learning, such that the trained policy is robust and safe against any other attackers. We can easily show that performing adversarial training with the MC or the MR attacker will enable the agent to be robust against the most effective or the most reward stealthy perturbations, respectively (see Appendix A.6 for details).

*Remark* 1. Suppose a trained policy $\pi'$ under the MC attacker satisfies: $V_c^{\pi' \circ \nu_{\text{MC}}}(\mu_0) \leq \kappa$, then $\pi' \circ \nu$ is guaranteed to be feasible with any $B_p^\epsilon$ bounded adversarial perturbations. Similarly, suppose a trained policy $\pi'$ under the MR attacker satisfies: $V_c^{\pi' \circ \nu_{\text{MR}}}(\mu_0) \leq \kappa$, then $\pi' \circ \nu$ is guaranteed to be non-tempting with any $B_p^\epsilon$ bounded adversarial perturbations.

Remark 1 indicates that by solving the adversarial constrained optimization problem under the MC attacker, all the feasible solutions will be safe under any bounded adversarial perturbations. It also shows a nice property for training a robust policy, since the max operation over the reward in the safe RL objective may lead the policy to the tempting policy class, while the adversarial training with MR attacker can naturally keep the trained policy at a safe distance from the tempting policy class. Practically, we observe that both MC and MR attackers can increase the robustness and safety via adversarial training, and could be easily plugged into any on-policy safe RL algorithms, in principle. We leave the robust training framework for off-policy safe RL methods as future work.

## 4.2 PRACTICAL IMPLEMENTATION

The meta adversarial training algorithm is shown in Algo. 1. We particularly adopt the primal-dual methods (Ray et al., 2019; Stooke et al., 2020) that are widely used in the safe RL literature as the `learner`, then the adversarial training objective in Eq. (5) can be converted to a min-max form by using the Lagrange multiplier $\lambda$:

---

**Algorithm 1** Adversarial safe RL training meta algorithm

**Input:** Safe RL `learner`, Adversary `scheduler`
**Output:** Observational robust policy $\pi$

1: Initialize policy $\pi \in \Pi$ and adversary $\nu : \mathcal{S} \to \mathcal{S}$
2: **for** each training epoch $n = 1, ..., N$ **do**
3:     Rollout trajectories: $\tilde{\tau} = \{s_0, \tilde{a}_0, ...\}_T$, $\tilde{a}_t \sim \pi(a|\nu(s_t))$
4:     Run safe RL learner: $\pi \leftarrow$ `learner`$(\tilde{\tau}, \Pi)$
5:     Update adversary: $\nu \leftarrow$ `scheduler`$(\tilde{\tau}, \pi, n)$
6: **end for**

---

$$(\pi^*, \lambda^*) = \min_{\lambda \geq 0} \max_{\pi \in \Pi} V_r^{\pi \circ \nu}(\mu_0) - \lambda(V_c^{\pi \circ \nu}(\mu_0) - \kappa) \tag{7}$$

Solving the inner maximization (primal update) via any policy optimization methods and the outer minimization (dual update) via gradient descent iteratively yields the Lagrangian algorithm. Under proper learning rates and bounded noise assumptions, the iterates $(\pi_n, \lambda_n)$ converge to a fixed point (a local minimum) almost surely (Tessler et al., 2018; Paternain et al., 2019).

Based on previous theoretical analysis, we adopt MC or MR as the adversary when sampling trajectories. The `scheduler` aims to train the reward and cost Q-value functions for the MR and the MC attackers, because many on-policy algorithms such as PPO do not use them. In addition, the scheduler can update the power of the adversary based on the learning progress accordingly, since a strong adversary at the beginning may prohibit the `learner` from exploring the environment and thus corrupt the training. We gradually increase the perturbation range $\epsilon$ along with the training epochs to adjust the adversary perturbation set $B_p^\epsilon$, such that the agent will not be too conservative in the early stage of training. A similar idea is also used in adversarial training (Salimans et al., 2016; Arjovsky & Bottou, 2017; Gowal et al., 2018) and curriculum learning literature (Dennis et al., 2020; Portelas et al., 2020). See more implementation details in Appendix C.3.

## 5 EXPERIMENT

In this section, we aim to answer the questions raised in Sec. 1. To this end, we adopt the robot locomotion continuous control tasks that are easy to interpret, motivated by safety, and used in many previous works (Achiam et al., 2017; Chow et al., 2019; Zhang et al., 2020b). The simulation environments are from a public available benchmark (Gronauer, 2022). We consider two tasks, and train multiple different robots (Car, Drone, Ant) for each task:

**Run task.** Agents are rewarded for running fast between two safety boundaries and are given costs for violation constraints if they run across the boundaries or exceed an agent-specific velocity threshold. The tempting policies can violate the velocity constraint to obtain more rewards.

**Circle task.** The agents are rewarded for running in a circle in a clockwise direction but are constrained to stay within a safe region that is smaller than the radius of the target circle. The tempting policies in this task will leave the safe region to gain more rewards.

We name each task via the `Robot-Task` format, for instance, `Car-Run`. More detailed descriptions and video demos are available on our anonymous project website [1]. In addition, we will use the PID PPO-Lagrangian (abbreviated as PPOL) method (Stooke et al., 2020) as the base safe RL algorithm to fairly compare different robust training approaches, while the proposed adversarial training can be easily used in other on-policy safe RL methods as well. The detailed hyperparameters of the adversaries and safe RL algorithms can be found in Appendix C.

### 5.1 ADVERSARIAL ATTACKER COMPARISON

We first demonstrate the vulnerability of the optimal safe RL policies without adversarial training and compare the performance of different adversaries. All the adversaries have the same $\ell_\infty$ norm perturbation set $B_\infty^\epsilon$ restriction. We adopt three adversary baselines, including one improved version:

**Random attacker baseline.** This is a simple baseline by sampling the corrupted observations randomly within the perturbation set via a uniform distribution.

**Maximum Action Difference (MAD) attacker baseline.** The MAD attacker (Zhang et al., 2020a) is designed for standard RL tasks, which is shown to be effective in decreasing a trained RL agent's reward return. The optimal adversarial observation is obtained by maximizing the KL-divergence between the corrupted policy: $\nu_{\text{MAD}}(s) = \arg\max_{\tilde{s} \in B_p^\epsilon(s)} D_{\text{KL}}\left[\pi(a|\tilde{s}) \| \pi(a|s)\right]$

**Adaptive MAD (AMAD) attacker.** Since the vanilla MAD attacker is not designed for safe RL, we further improve it to an adaptive version as a stronger baseline. The motivation comes from Lemma 2 – the optimal policy will be close to the constraint boundary that with high risks. To better understand this property, we introduce the discounted future state distribution $d^\pi(s)$ (Kakade, 2003), which allows us to rewrite the result in Lemma 2 as (see Appendix C.6 for derivation and implementation details): $\frac{1}{1-\gamma} \int_{s \in \mathcal{S}} d^{\pi^*}(s) \int_{a \in \mathcal{A}} \pi^*(a|s) \int_{s' \in \mathcal{S}} p(s'|s,a)c(s,a,s')ds'dads = \kappa$. We can see that performing MAD attack for the optimal policy $\pi^*$ in low-risk regions that with small $p(s'|s,a)c(s,a,s')$ values may not be effective. Therefore, AMAD only perturbs the observation when the agent is within high-risk regions that are determined by the cost value function and a threshold $\xi$ to achieve more effective attacks: $\nu_{\text{AMAD}}(s) := \begin{cases} \nu_{\text{MAD}}(s), & \text{if } V_c^\pi(s) \geq \xi, \\ s, & \text{otherwise .} \end{cases}$

**Experiment setting.** We evaluate the performance of all three baselines above and our MC, MR adversaries by attacking well-trained PPO-Lagrangian policies in different tasks. The trained policies can achieve nearly zero constraint violation costs without observational perturbations. We keep the trained model weights and environment seeds fixed for all the attackers to ensure fair comparisons.

**Experiment result.** Fig. 2 shows the attack results of the 5 adversaries on PPOL-vanilla. Each column corresponds to an environment. The first row is the episode reward and the second row is the episode cost of constraint violations. We can see that the vanilla safe RL policies are vulnerable, since the safety performance deteriorates (cost increases) significantly even with a small adversarial perturbation range $\epsilon$. Generally, we can see an increasing cost trend as the $\epsilon$ increases, except for the MAD attacker. Although MAD can reduce the agent's reward quite well, it fails to perform an effective attack in increasing the cost because the reward decrease may keep the agent away

---
[1]`https://sites.google.com/view/robustsaferl/home`

from high-risk regions. It is even worse than the random attacker in the `Car-Circle` task. The improved AMAD attacker is a stronger baseline than MAD, as it only attacks in high-risk regions and thus has a higher chance of entering unsafe regions to induce more constraint violations. More comparisons between MAD and AMAD can be found in Appendix C.9. Our proposed MC and MR attackers outperform all baselines attackers (Random, MAD and AMAD) in terms of effectiveness by increasing the cost by a large margin in most tasks. Surprisingly, the MR attacker can achieve even higher costs than MC and is more stealthy as it can maintain or increase the reward well, which validates our theoretical analysis and the existence of tempting policies.

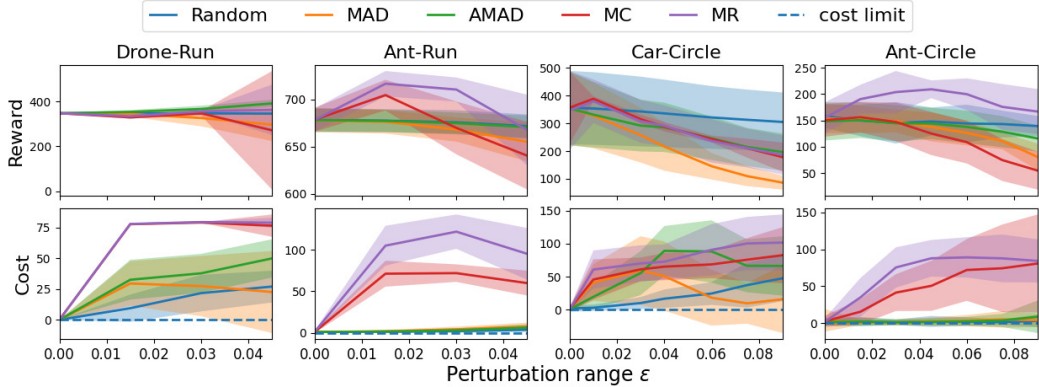

Figure 2: Reward and cost curves of all 5 attackers evaluated on well-trained vanilla PPO-Lagrangian models w.r.t. the perturbation range $\epsilon$. The curves are averaged over 50 episodes and 5 seeds, where the solid lines are the mean and the shadowed areas are the standard deviation. The dashed line is the cost without perturbations.

## 5.2 PERFORMANCE OF SAFE RL WITH ADVERSARIAL TRAINING

We adopt 5 baselines, including the **PPOL-vanilla** method without robust training, the naive adversarial training under random noise **PPOL-random**, the state-adversarial algorithm **SA-PPOL** proposed in (Zhang et al., 2020a), but we extend their PPO in standard RL setting to PPOL in the safe RL setting. The original SA-PPOL algorithm utilizes the MAD attacker to compute the adversarial observations, and then adds a KL regularizer to penalize the divergence between them and the original observations. We add two additional baselines **SA-PPOL(MC)** and **SA-PPOL(MR)** for the ablation study, where we change the MAD attacker to our proposed MC and MR adversaries. Our adversarial training methods are named as **ADV-PPOL(MC)** and **ADV-PPOL(MR)**, which are trained under the MC and MR attackers respectively. We use the same PPOL implementation and hyperparameters for all methods for fair comparisons. More details can be found in Appendix C.5-C.8.

**Results.** The evaluation results of different trained policies under adversarial attacks are shown in Table 1, where **Natural** represents the performance without noise. We train each algorithm with 5 random seeds and evaluate each trained policy with 50 episodes under each attacker to obtain the values. The training and testing perturbation range $\epsilon$ is the same. We use gray shadows to highlight the top two safest agents with the smallest cost values, but we ignore the failure agents whose rewards are less than 30% of the PPOL-vanilla method. We mark the failure agents with $\star$. Due to the page limit, we leave the evaluation results under random and MAD attackers to Appendix C.9.

**Analysis.** We can observe that although most baselines can achieve near zero natural cost, their safety performances are vulnerable under the strong MC and MR attackers, which are more effective than AMAD in inducing unsafe behaviors. The proposed adversarial training methods (ADV-PPOL) consistently outperform baselines in safety with the lowest costs while maintaining high rewards in most tasks. The comparison with PPOL-random indicates that the MC and MR attackers are essential ingredients of adversarial training. Although SA-PPOL agents can maintain reward very well, they are not safe as to constraint satisfaction under adversarial perturbations in most environments. The ablation studies with SA-PPOL(MC) and SA-PPOL(MR) suggest that the KL-regularized robust training technique, which is successful in standard robust RL setting, does not work well for safe RL even with the same adversarial attacks during training, and they may also fail to obtain a high-rewarding policy in some tasks (see discussions of the training failure in Appendix B.2). As a result, we can conclude that the proposed adversarial training methods with the MC and MR attackers are better than baselines regarding both training stability and testing robustness and safety.

Table 1: Evaluation results of natural performance (no attack) and under 3 attackers. Our methods are ADV-PPOL(MC/MR). Each value is reported as: mean ± standard deviation for 50 episodes and 5 seeds. We shadow two lowest-costs agents under each attacker column and break ties based on rewards, excluding the failing agents (whose natural rewards are less than 30% of PPOL-vanilla's). We mark the failing agents with $\star$.

| Env | Method | Natural | | AMAD | | MC | | MR | |
|---|---|---|---|---|---|---|---|---|---|
| | | Reward | Cost | Reward | Cost | Reward | Cost | Reward | Cost |
| Car-Run $\epsilon = 0.05$ | PPOL-vanilla | 560.86±1.09 | 0.16±0.36 | 559.45±2.87 | 3.7±7.65 | 624.92±16.22 | 184.04±0.67 | 625.12±15.96 | 184.08±0.46 |
| | PPOL-random | 557.27±1.06 | 0.0±0.0 | 556.46±1.07 | 0.28±0.71 | 583.52±1.59 | 183.78±0.7 | 583.43±1.46 | 183.88±0.55 |
| | SA-PPOL | 534.3±8.84 | 0.0±0.0 | 534.22±8.91 | 0.0±0.0 | 566.75±7.68 | 13.77±13.06 | 566.54±7.4 | 11.79±11.95 |
| | SA-PPOL(MC) | 552.0±2.76 | 0.0±0.0 | 550.68±2.81 | 0.0±0.0 | 568.28±3.98 | 3.28±5.73 | 568.93±3.17 | 2.73±5.27 |
| | SA-PPOL(MR) | 548.71±2.03 | 0.0±0.0 | 547.61±1.93 | 0.0±0.0 | 568.49±5.2 | 25.72±51.51 | 568.72±4.92 | 24.33±48.74 |
| | ADV-PPOL(MC) | 505.76±9.11 | 0.0±0.0 | 503.49±9.17 | 0.0±0.0 | 552.98±3.76 | 0.0±0.06 | 549.07±8.22 | 0.02±0.14 |
| | ADV-PPOL(MR) | 497.67±8.15 | 0.0±0.0 | 494.81±7.49 | 0.0±0.0 | 549.24±9.98 | 0.02±0.15 | 551.75±7.63 | 0.04±0.21 |
| Drone-Run $\epsilon = 0.025$ | PPOL-vanilla | 346.1±2.71 | 0.0±0.0 | 344.95±3.08 | 1.76±4.15 | 339.62±5.12 | 79.0±0.0 | 359.01±14.62 | 78.82±0.43 |
| | PPOL-random | 342.66±0.96 | 0.0±0.0 | 357.56±19.31 | 31.36±35.64 | 265.42±3.08 | 0.04±0.57 | 317.26±29.93 | 33.31±19.26 |
| | SA-PPOL | 338.5±2.26 | 0.0±0.0 | 358.66±32.06 | 33.27±34.58 | 313.81±163.22 | 52.44±28.28 | 264.08±168.62 | 42.8±22.61 |
| | SA-PPOL(MC) | 223.1±22.5 | 0.84±1.93 | 210.61±28.78 | 0.82±1.88 | 251.67±31.72 | 22.98±16.69 | 262.73±29.1 | 21.48±16.18 |
| | *SA-PPOL(MR) | 0.3±0.49 | 0.0±0.0 | 0.3±0.45 | 0.0±0.0 | 0.44±0.87 | 0.0±0.0 | 0.17±0.43 | 0.0±0.0 |
| | ADV-PPOL(MC) | 263.24±9.67 | 0.0±0.0 | 268.66±15.34 | 0.0±0.0 | 272.34±52.35 | 3.0±6.5 | 282.36±39.84 | 13.48±13.8 |
| | ADV-PPOL(MR) | 226.18±74.06 | 0.0±0.0 | 225.34±75.01 | 0.0±0.0 | 227.89±61.5 | 3.58±7.44 | 242.47±80.6 | 6.62±8.84 |
| Ant-Run $\epsilon = 0.025$ | PPOL-vanilla | 703.11±3.83 | 1.3±1.17 | 702.31±3.76 | 2.53±1.71 | 692.88±9.32 | 65.56±9.56 | 714.37±26.4 | 120.68±28.63 |
| | PPOL-random | 698.39±14.76 | 1.34±1.39 | 697.56±14.38 | 2.02±1.47 | 648.88±83.55 | 54.52±24.27 | 677.95±52.34 | 80.96±42.04 |
| | SA-PPOL | 699.7±12.1 | 0.66±0.82 | 699.48±12.19 | 1.02±1.11 | 683.03±21.1 | 70.54±27.69 | 723.52±36.33 | 122.69±39.75 |
| | SA-PPOL(MC) | 383.21±256.58 | 5.71±6.34 | 382.32±256.39 | 5.46±6.03 | 402.83±274.66 | 34.28±42.18 | 406.31±276.04 | 38.5±46.93 |
| | *SA-PPOL(MR) | 114.34±35.83 | 6.63±3.7 | 115.3±35.13 | 6.55±3.72 | 112.7±32.76 | 9.64±3.76 | 115.77±33.51 | 9.6±3.72 |
| | ADV-PPOL(MC) | 615.4±2.94 | 0.0±0.0 | 614.96±2.94 | 0.0±0.06 | 674.65±12.01 | 2.21±1.64 | 675.87±20.64 | 5.3±3.11 |
| | ADV-PPOL(MR) | 596.14±12.06 | 0.0±0.0 | 595.52±12.03 | 0.0±0.0 | 657.31±17.09 | 0.96±1.11 | 678.65±13.16 | 1.56±1.41 |
| Car-Circle $\epsilon = 0.05$ | PPOL-vanilla | 446.83±9.89 | 1.32±3.61 | 406.75±15.82 | 21.85±24.9 | 248.05±21.66 | 38.56±24.01 | 296.17±20.95 | 89.23±17.11 |
| | PPOL-random | 429.57±10.55 | 0.06±1.01 | 442.89±11.26 | 41.85±12.06 | 289.17±30.67 | 70.9±23.24 | 313.31±25.77 | 95.23±13.62 |
| | SA-PPOL | 435.83±10.98 | 0.34±1.55 | 430.58±10.41 | 7.48±15.43 | 295.38±88.05 | 126.3±33.87 | 468.74±12.4 | 94.19±11.62 |
| | SA-PPOL(MC) | 439.18±10.12 | 0.27±1.27 | 352.71±53.84 | 0.1±0.45 | 311.04±41.29 | 91.07±16.8 | 450.93±20.37 | 83.89±15.59 |
| | SA-PPOL(MR) | 419.9±34.0 | 0.32±1.29 | 411.32±36.23 | 0.31±1.04 | 317.01±72.81 | 99.3±22.89 | 421.31±67.83 | 83.89±15.59 |
| | ADV-PPOL(MC) | 270.25±16.99 | 0.0±0.0 | 273.48±17.52 | 0.0±0.0 | 263.5±24.5 | 1.44±3.48 | 248.43±40.74 | 8.99±7.46 |
| | ADV-PPOL(MR) | 274.69±20.5 | 0.0±0.0 | 281.73±21.43 | 0.0±0.0 | 219.29±31.25 | 2.21±5.7 | 281.12±25.89 | 1.66±2.52 |
| Drone-Circle $\epsilon = 0.025$ | PPOL-vanilla | 706.94±53.66 | 4.55±6.58 | 634.54±129.07 | 29.04±17.75 | 153.18±147.23 | 24.14±30.25 | 121.85±159.92 | 20.01±29.08 |
| | PPOL-random | 728.62±64.07 | 1.2±3.75 | 660.72±122.6 | 28.3±19.42 | 194.63±149.35 | 12.9±18.63 | 165.13±165.01 | 23.3±24.58 |
| | SA-PPOL | 599.56±67.56 | 1.71±3.39 | 596.98±67.66 | 1.93±3.81 | 338.85±204.86 | 72.83±43.65 | 84.2±132.76 | 20.43±31.26 |
| | SA-PPOL(MC) | 480.34±96.61 | 2.7±4.12 | 475.21±97.68 | 1.25±3.44 | 361.46±190.63 | 54.71±39.13 | 248.74±203.66 | 36.68±32.19 |
| | SA-PPOL(MR) | 335.99±150.18 | 2.8±5.55 | 326.73±152.64 | 2.66±4.85 | 233.8±158.16 | 51.79±38.98 | 287.92±194.92 | 52.39±41.26 |
| | ADV-PPOL(MC) | 309.83±64.1 | 0.0±0.0 | 279.91±85.93 | 4.25±8.62 | 393.66±92.91 | 0.88±2.65 | 250.59±112.6 | 11.16±22.11 |
| | ADV-PPOL(MR) | 358.23±40.59 | 0.46±2.35 | 360.4±42.24 | 0.4±3.9 | 289.1±90.7 | 6.77±9.58 | 363.75±74.02 | 2.44±5.2 |
| Ant-Circle $\epsilon = 0.025$ | PPOL-vanilla | 186.71±28.65 | 4.47±7.22 | 185.15±25.72 | 5.26±8.65 | 185.89±34.57 | 67.43±24.58 | 232.42±37.32 | 80.59±20.41 |
| | PPOL-random | 140.1±25.56 | 3.58±7.6 | 143.25±17.97 | 4.22±8.21 | 139.42±27.53 | 35.69±26.59 | 155.77±32.44 | 54.54±28.12 |
| | SA-PPOL | 197.9±27.39 | 3.4±8.04 | 196.2±32.59 | 4.06±8.93 | 198.73±32.08 | 76.45±27.26 | 246.8±40.61 | 82.24±20.28 |
| | *SA-PPOL(MC) | 0.65±0.43 | 0.0±0.0 | 0.66±0.43 | 0.0±0.0 | 0.63±0.42 | 0.0±0.0 | 0.63±0.38 | 0.0±0.0 |
| | *SA-PPOL(MR) | 0.63±0.41 | 0.0±0.0 | 0.63±0.41 | 0.0±0.0 | 0.58±0.44 | 0.0±0.0 | 0.64±0.44 | 0.0±0.0 |
| | ADV-PPOL(MC) | 121.57±20.11 | 1.24±4.7 | 122.2±20.55 | 0.98±4.43 | 124.29±26.04 | 1.9±5.28 | 107.89±21.35 | 9.0±17.31 |
| | ADV-PPOL(MR) | 123.13±19.19 | 0.46±2.69 | 121.51±19.68 | 0.74±3.42 | 110.11±25.49 | 5.72±10.1 | 128.88±20.06 | 3.0±7.9 |

**Generalization to other safe RL methods.** We also conduct the experiments for other types of base safe RL algorithms, including another on-policy method FOCOPS (Zhang et al., 2020b), one off-policy method SAC-Lagrangian Yang et al. (2021), and one policy-gradient-free off-policy method CVPO (Liu et al., 2022). Due to the page limit, we leave the results and detailed discussions in Appendix C.9. In summary, all the vanilla safe RL methods suffer the vulnerability issue – though they are safe in noise-free environments, they are not safe anymore under strong attacks, which validates the necessity of studying the observational robustness of safe RL agents. In addition, the adversarial training can help to improve the robustness and make the FOCOPS agent much safer under attacks. Therefore, the problem formulations, methods, results, and analysis can be generalized to different safe RL approaches, hopefully attracting more attention in the safe RL community to study the inherent connection between safety and robustness.

## 6 CONCLUSION

We study the observational robustness regarding constraint satisfaction for safe RL and show that the optimal policy of tempting problems could be vulnerable. We propose two effective attackers to induce unsafe behaviors. An interesting and surprising finding is that maximizing-reward attack is as effective as directly maximizing the cost while keeping stealthiness. We further propose an adversarial training method to increase the robustness and safety performance, and extensive experiments show that the proposed method outperforms the robust training techniques for standard RL settings.

One limitation of this work is that the adversarial training pipeline could be expensive for real-world RL applications because it requires to attack the behavior agents when collecting data. In addition, the adversarial training might be unstable for high-dimensional and complex problems. Nevertheless, our results show the existence of a previously unrecognized problem in safe RL, and we hope this work encourages other researchers to study safety from the robustness perspective, as both safety and robustness are important ingredients for real-world deployment.

ACKNOWLEDGMENTS

We gratefully acknowledge support from the National Science Foundation under grant CAREER CNS-2047454.

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

# Table of Contents

# A  PROOFS AND DISCUSSIONS

## A.1  PROOF OF LEMMA 1 AND PROPOSITION 1 – INFEASIBLE TEMPTING POLICIES

Lemma 1 indicates that all the tempting policies are infeasible: $\forall \pi \in \Pi_{\mathcal{M}}^T, V_c^\pi(\mu_0) > \kappa$. We will prove it by contradiction.

*Proof.* For a tempting safe RL problem $\mathcal{M}_\Pi^\kappa$, there exists a tempting policy that satisfies the constraint: $\pi' \in \Pi_{\mathcal{M}}^T, V_c^{\pi'}(\mu_0) \leq \kappa, \pi' \in \Pi_{\mathcal{M}}^\kappa$. Denote the optimal policy as $\pi^*$, then based on the definition of the tempting policy, we have $V_r^{\pi'}(\mu_0) > V_r^{\pi^*}(\mu_0)$. Based on the definition of optimality, we know that for any other feasible policy $\pi \in \Pi_{\mathcal{M}}^\kappa$, we have:

$$V_r^{\pi'}(\mu_0) > V_r^{\pi^*}(\mu_0) \geq V_r^\pi(\mu_0),$$

which indicates that $\pi'$ is the optimal policy for $\mathcal{M}_\Pi^\kappa$. Then again, based on the definition of tempting policy, we will obtain:

$$V_r^{\pi'}(\mu_0) > V_r^{\pi'}(\mu_0),$$

which contradicts to the fact that $V_r^{\pi'}(\mu_0) = V_r^{\pi'}(\mu_0)$. Therefore, there is no tempting policy that satisfies the constraint. $\square$

Proposition 1 suggest that as long as the MR attacker can successfully obtain a policy that has higher reward return than the optimal policy $\pi^*$ given enough large perturbation set $B_p^\epsilon(s)$, it is guaranteed to be reward stealthy and effective.

*Proof.* The stealthiness is naturally satisfied based on the definition. The effectiveness is guaranteed by Lemma 1. Since the corrupted policy $\pi^* \circ \nu_{\text{MR}}$ can achieve $V_r^{\pi^* \circ \nu_{\text{MR}}} > V_r^{\pi^*}$, we can conclude that $\pi^* \circ \nu_{\text{MR}}$ is within the tempting policy class, since it has higher reward than the optimal policy. Then we know that it will violate the constraint based on Lemma 1, and thus the MR attacker is effective. $\square$

## A.2   PROOF OF LEMMA 2 – OPTIMAL POLICY'S COST VALUE

Lemma 2 says that in augmented policy space $\bar{\Pi}$, the optimal policy $\pi^*$ of a tempting safe RL problem satisfies: $V_c^{\pi^*}(\mu_0) = \kappa$. It is clear to see that the temping policy space and the original policy space are subsets of the augmented policy space: $\Pi_{\mathcal{M}}^T \subset \Pi \subset \bar{\Pi}$. We then prove Lemma 2 by contradiction.

*Proof.* Suppose the optimal policy $\pi^*(a|s,o)$ in augmented policy space for a tempting safe RL problem has $V_c^{\pi^*}(\mu_0) < \kappa$ and its option update function is $\pi_o^*$. Denote $\pi' \in \Pi_{\mathcal{M}}^T$ as a tempting policy. Based on Lemma 1, we know that $V_c^{\pi'}(\mu_0) > \kappa$ and $V_r^{\pi'}(\mu_0) > V_r^{\pi^*}(\mu_0)$. Then we can compute a weight $\alpha$:

$$\alpha = \frac{\kappa - V_c^{\pi^*}(\mu_0)}{V_c^{\pi'}(\mu_0) - V_c^{\pi^*}(\mu_0)}. \tag{8}$$

We can see that:

$$\alpha V_c^{\pi'}(\mu_0) + (1-\alpha)V_c^{\pi^*}(\mu_0) = \kappa. \tag{9}$$

Now we consider the augmented space $\bar{\Pi}$. Since $\Pi \subseteq \bar{\Pi}$, $\pi^*, \pi' \in \bar{\Pi}$, and then we further define another policy $\bar{\pi}$ based on the trajectory-wise mixture of $\pi^*$ and $\pi'$ as

$$\bar{\pi}(a_t|s_t, o_t) = \begin{cases} \pi'(a_t|s_t), & \text{if } o_t = 1 \\ \pi^*(a_t|s_t, u_t), & \text{if } o_t = 0 \end{cases} \tag{10}$$

with $o_{t+1} = o_t, o_0 \sim \text{Bernoulli}(\alpha)$ and the update of $u$ follows the definition of $\pi_o^*$. Therefore, the trajectory of $\bar{\pi}$ has $\alpha$ probability to be sampled from $\pi'$ and $1-\alpha$ probability to be sampled from $\pi^*$:

$$\tau \sim \bar{\pi} := \begin{cases} \tau \sim \pi', & \text{with probability } \alpha, \\ \tau \sim \pi^*, & \text{with probability } 1-\alpha. \end{cases} \tag{11}$$

Then we can conclude that $\bar{\pi}$ is also feasible:

$$V_c^{\bar{\pi}}(\mu_0) = \mathbb{E}_{\tau \sim \bar{\pi}}[\sum_{t=0}^{\infty} \gamma^t c_t] = \alpha\mathbb{E}_{\tau \sim \pi'}[\sum_{t=0}^{\infty} \gamma^t c_t] + (1-\alpha)\mathbb{E}_{\tau \sim \pi^*}[\sum_{t=0}^{\infty} \gamma^t c_t] \tag{12}$$

$$= \alpha V_c^{\pi'}(\mu_0) + (1-\alpha)V_c^{\pi^*}(\mu_0) = \kappa. \tag{13}$$

In addition, $\bar{\pi}$ has higher reward return than the optimal policy $\pi^*$:

$$V_r^{\bar{\pi}}(\mu_0) = \mathbb{E}_{\tau \sim \bar{\pi}}[\sum_{t=0}^{\infty} \gamma^t r_t] = \alpha\mathbb{E}_{\tau \sim \pi'}[\sum_{t=0}^{\infty} \gamma^t r_t] + (1-\alpha)\mathbb{E}_{\tau \sim \pi^*}[\sum_{t=0}^{\infty} \gamma^t r_t] \tag{14}$$

$$= \alpha V_r^{\pi'}(\mu_0) + (1-\alpha)V_r^{\pi^*}(\mu_0) \tag{15}$$

$$> \alpha V_r^{\pi^*}(\mu_0) + (1-\alpha)V_r^{\pi^*}(\mu_0) = V_r^{\pi^*}(\mu_0), \tag{16}$$

where the inequality comes from the definition of the tempting policy. Since $\bar{\pi}$ is both feasible, and has strictly higher reward return than the policy $\pi^*$, we know that $\pi^*$ is not optimal, which contradicts to our assumption. Therefore, the optimal policy $\pi^*$ should always satisfy $V_c^{\pi^*}(\mu_0) = \kappa$.

$\square$

*Remark* 2. The cost value function $V_c^{\pi^*}(\mu_0) = \mathbb{E}_{\tau \sim \pi}[\sum_{t=0}^{\infty} \gamma^t c_t]$ is based on the expectation of the sampled trajectories (expectation over episodes) rather than a single trajectory (expectation within one episode), because for a single sampled trajectory $\tau \sim \pi$, $V_c^{\pi^*}(\tau) = \sum_{t=0}^{\infty} \gamma^t c_t$ may even not necessarily satisfy the constraint.

*Remark* 3. The proof also indicates that the range of metric function $\mathcal{V} := \{(V_r^{\pi}(\mu_0), V_c^{\pi}(\mu_0))\}$ (as shown as the blue circle in Fig.1) is convex when we extend $\bar{\Pi}$ to a linear mixture of $\Pi$, i.e., let $\mathcal{O} = \{1, 2, 3, \dots\}$ and $\bar{\Pi} : \mathcal{S} \times \mathcal{A} \times \mathcal{O} \to [0, 1]$. Consider $\boldsymbol{\alpha} = [\alpha_1, \alpha_2, \dots], \alpha_i \geq 0, \sum_{i=1} \alpha_i = 1, \boldsymbol{\pi} = [\pi_1, \pi_2, \dots]$. We can construct a policy $\bar{\pi} \in \bar{\Pi} = \langle \boldsymbol{\alpha}, \boldsymbol{\pi} \rangle$:

$$\bar{\pi}(a_t|s_t, o_t) = \begin{cases} \pi_1(a_t|s_t), & \text{if } o_t = 1 \\ \pi_2(a_t|s_t), & \text{if } o_t = 2 \\ \dots \end{cases} \tag{17}$$

with $o_{t+1} = o_t, \Pr(o_0 = i) = \alpha_i$. Then we have

$$\tau \sim \langle \boldsymbol{\alpha}, \boldsymbol{\pi} \rangle := \tau \sim \pi_i, \text{ with probability } \alpha_i, i = 1, 2, \ldots, \tag{18}$$

Similar to the above proof, we have

$$V_f^{\langle \boldsymbol{\alpha}, \boldsymbol{\pi} \rangle}(\mu_0) = \langle \boldsymbol{\alpha}, V_f^{\boldsymbol{\pi}}(\mu_0) \rangle, f \in \{r, c\}, \tag{19}$$

where $V_f^{\boldsymbol{\pi}}(\mu_0) = [V_f^{\pi_1}(\mu_0), V_f^{\pi_2}(\mu_0), \ldots]$. Consider $\forall (v_{r1}, v_{c1}), (v_{r2}, v_{c2}) \in \mathcal{V}$, suppose they correspond to policy mixture $\langle \boldsymbol{\alpha}, \boldsymbol{\pi} \rangle$ and $\langle \boldsymbol{\beta}, \boldsymbol{\pi} \rangle$ respectively, then $\forall t \in [0, 1]$, the new mixture $\langle t\boldsymbol{\alpha} + (1-t)\boldsymbol{\beta}, \boldsymbol{\pi} \rangle \in \bar{\Pi}$ and $V_f^{\langle t\boldsymbol{\alpha} + (1-t)\boldsymbol{\beta}, \boldsymbol{\pi} \rangle}(\mu_0) = t \cdot v_{f1} + (1-t) \cdot v_{f2} \in \mathcal{V}$. Therefore, $\mathcal{V}$ is a convex set.

*Remark* 4. The enlarged policy space guarantees the validation of Lemma 2 but is not always indispensable. In most environments, with the original policy space $\Pi$, the metric function space $\mathcal{V} := \{(V_r^\pi(\mu_0), V_c^\pi(\mu_0)) \mid \pi \in \Pi\}$ is a connected set (i.e., there exists a $\bar{\pi} \in \Pi$ such that $V_c^{\bar{\pi}}(\mu_0) = \kappa$ if there are $\pi_1, \pi_2 \in \Pi, s.t. V_c^{\pi_1}(\mu_0) < \kappa < V_c^{\pi_2}(\mu_0)$) and we can obtain the optimal policy exactly on the constraint boundary without expanding policy space.

### A.3 PROOF OF THEOREM 1 – EXISTENCE OF OPTIMAL DETERMINISTIC MC/MR ADVERSARY

**Existence**. Given a fixed policy $\pi$, We first introduce two adversary MDPs $\hat{\mathcal{M}}_r = (\mathcal{S}, \hat{\mathcal{A}}, \hat{\mathcal{P}}, \hat{R}_r, \gamma)$ for reward maximization adversary and $\hat{\mathcal{M}}_c = (\mathcal{S}, \hat{\mathcal{A}}, \hat{\mathcal{P}}, \hat{R}_c, \gamma)$ for cost maximization adversary to prove the existence of optimal adversary. In adversary MDPs, the adversary acts as the agent to choose a perturbed state as the action (i.e., $\hat{a} = \tilde{s}$) to maximize the cumulative reward $\sum \hat{R}$. Therefore, in adversary MDPs, the action space $\hat{\mathcal{A}} = \mathcal{S}$ and $\nu(\cdot|s)$ denotes a policy distribution.

Based on the above definitions, we can also derive transition function and reward function for new MDPs Zhang et al. (2020a)

$$\hat{p}(s'|s, a) = \sum_a \pi(a|\hat{a})p(s'|s, a), \tag{20}$$

$$\hat{R}_f(s, \hat{a}, s') = \begin{cases} \frac{\sum_a \pi(a|\hat{a})p(s'|s,a)f(s,a,s')}{\sum_a \pi(a|\hat{a})p(s'|s,a)}, & \hat{a} \in B_p^\epsilon(s) \\ -C, & \hat{a} \notin B_p^\epsilon(s) \end{cases}, f \in \{r, c\}, \tag{21}$$

where $\hat{a} = \tilde{s} \sim \nu(\cdot|s)$ and $C$ is a constant. Therefore, with sufficiently large $C$, we can guarantee that the optimal adversary $\nu^*$ will not choose a perturbed state $\hat{a}$ out of the $l_p$-ball of the given state $s$, i.e., $\nu^*(\hat{a}|s) = 0, \forall \hat{a} \notin B_p^\epsilon(s)$.

According to the properties of MDP Sutton et al. (1998), $\hat{\mathcal{M}}_r, \hat{\mathcal{M}}_c$ have corresponding optimal policy $\nu_r^*, \nu_c^*$, which are deterministic by assigning unit mass probability to the optimal action $\hat{a}$ for each state.

Next, we will prove that $\nu_r^* = \nu_{\text{MR}}, \nu_c^* = \nu_{\text{MC}}$. Consider value function in $\hat{\mathcal{M}}_f, f \in \{r, c\}$, for an adversary $\nu \in \mathcal{N} := \{\nu | \nu^*(\hat{a}|s) = 0, \forall \hat{a} \notin B_p^\epsilon(s)\}$, we have

$$\hat{V}_f^\nu(s) = \mathbb{E}_{\hat{a} \sim \nu(\cdot|s), s' \sim \hat{p}(\cdot|s, \hat{a})}[\hat{R}_f(s, \hat{a}, s') + \gamma \hat{V}_f^\nu(s')] \tag{22}$$

$$= \sum_{\hat{a}} \nu(\hat{a}|s) \sum_{s'} \hat{p}(s'|s, \hat{a})[\hat{R}_f(s, \hat{a}, s') + \gamma \hat{V}_f^\nu(s')] \tag{23}$$

$$= \sum_{\hat{a}} \nu(\hat{a}|s) \sum_{s'} \sum_a \pi(a|\hat{a})p(s'|s, a) \left[ \frac{\sum_a \pi(a|\hat{a})p(s'|s,a)f(s,a,s')}{\sum_a \pi(a|\hat{a})p(s'|s,a)} + \gamma \hat{V}_f^\nu(s') \right] \tag{24}$$

$$= \sum_{s'} p(s'|s, a) \sum_a \pi(a|\hat{a}) \sum_{\hat{a}} \nu(\hat{a}|s)[f(s, a, s') + \gamma \hat{V}_f^\nu(s')] \tag{25}$$

$$= \sum_{s'} p(s'|s, a) \sum_a \pi(a|\nu(s))[f(s, a, s') + \gamma \hat{V}_f^\nu(s')]. \tag{26}$$

Recall the value function in original safe RL problem,

$$V_f^{\pi \circ \nu}(s) = \sum_{s'} p(s'|s, a) \sum_a \pi(a|\nu(s))[f(s, a, s') + \gamma V_f^{\pi \circ \nu}(s')]. \tag{27}$$

Therefore, $V_f^{\pi \circ \nu}(s) = \hat{V}_f^{\nu}(s), \nu \in \mathcal{N}$. Note that in adversary MDPs $\nu_f^* \in \mathcal{N}$ and

$$\nu_f^* = \arg \max_{\nu} \mathbb{E}_{a \sim \pi(\cdot|\nu(s)), s' \sim p(\cdot|s,a)}[f(s,a,s') + \gamma \hat{V}_f^{\nu}(s')]. \tag{28}$$

We also know that $\nu_f^*$ is deterministic,

$$\Rightarrow \nu_f^*(s) = \arg \max_{\nu} \mathbb{E}_{a \sim \pi(\cdot|\tilde{s}), s' \sim p(\cdot|s,a)}[f(s,a,s') + \gamma \hat{V}_f^{\nu}(s')] \tag{29}$$

$$= \arg \max_{\nu} \mathbb{E}_{a \sim \pi(\cdot|\tilde{s}), s' \sim p(\cdot|s,a)}[f(s,a,s') + \gamma V_f^{\pi \circ \nu}(s')] \tag{30}$$

$$= \arg \max_{\nu} V_f^{\pi \circ \nu}(s,a). \tag{31}$$

Therefore, $\nu_r^* = \nu_{\mathrm{MR}}, \nu_c^* = \nu_{\mathrm{MC}}$.

**Optimality**. We will prove the optimality by contradiction. By definition, $\forall s \in \mathcal{S}$,

$$V_c^{\pi \circ \nu'}(s_0) \le V_c^{\pi \circ \nu_{\mathrm{MC}}}(s_0). \tag{32}$$

Suppose $\exists \nu', s.t. V_c^{\pi \circ \nu'}(\mu_0) > V_c^{\pi \circ \nu_{\mathrm{MC}}}(\mu_0)$, then there also exists $s_0 \in \mathcal{S}, s.t. V_c^{\pi \circ \nu'}(s_0) > V_c^{\pi \circ \nu_{\mathrm{MC}}}(s_0)$, which is contradictory to Eq.(32). Similarly, we can also prove that the property holds for $\nu_{\mathrm{MR}}$ by replacing $V_c^{\pi \circ \nu}$ with $V_r^{\pi \circ \nu}$. Therefore, there is no other adversary that achieves higher attack effectiveness than $\nu_{\mathrm{MR}}$ or higher reward stealthiness than $\nu_{\mathrm{MR}}$.

### A.4 PROOF OF THEOREM 2 – ONE-STEP ATTACK COST BOUND

We have

$$\tilde{V}_c^{\pi, \nu}(s) = \mathbb{E}_{a \sim \pi(\cdot|\nu(s)), s' \sim p(\cdot|s,a)}[c(s,a,s') + \gamma V_c^{\pi}(s')]. \tag{33}$$

By Bellman equation,

$$V_c^{\pi}(s) = \mathbb{E}_{a \sim \pi(\cdot|s), s' \sim p(\cdot|s,a)}[c(s,a,s') + \gamma V(s')]. \tag{34}$$

For simplicity, denote $p_{sa}^{s'} = p(s'|s,a)$ and we have

$$\tilde{V}_c^{\pi, \nu}(s) - V_c^{\pi}(s) = \sum_{a \in \mathcal{A}} \left( \pi(a|\nu(s)) - \pi(a|s) \sum_{s \in \mathcal{S}} p_{sa}^{s'} (c(s,a,s') + \gamma V_c^{\pi}(s')) \right) \tag{35}$$

$$\le \left( \sum_{a \in \mathcal{A}} |\pi(a|\nu(s)) - \pi(a|s)| \right) \max_{a \in \mathcal{A}} \sum_{s \in \mathcal{S}} p_{sa}^{s'} (c(s,a,s') + \gamma V_c^{\pi}(s')). \tag{36}$$

By definition, $D_{\mathrm{TV}}[\pi(\cdot|\nu(s)\|\pi(\cdot|s)] = \frac{1}{2}\sum_{a \in \mathcal{A}} |\pi(a|\nu(s)) - \pi(a|s)|$, and $c(s,a,s') = 0, s' \in \mathcal{S}_c$. Therefore, we have

$$\tilde{V}_c^{\pi, \nu}(s) - V_c^{\pi}(s) \le 2 D_{TV}[\pi(\cdot|\nu(s)\|\pi(\cdot|s)] \max_{a \in \mathcal{A}} \left( \sum_{s \in \mathcal{S}_c} p_{sa}^{s'} c(s,a,s') + \sum_{s \in \mathcal{S}} p_{sa}^{s'} \gamma V_c^{\pi}(s') \right) \tag{37}$$

$$\le 2L\|\nu(s) - s\|_p \max_{a \in \mathcal{A}} \left( \sum_{s \in \mathcal{S}_c} p_{sa}^{s'} C_m + \sum_{s \in \mathcal{S}} p_{sa}^{s'} \gamma \frac{C_m}{1 - \gamma} \right) \tag{38}$$

$$\le 2L\epsilon \left( p_s C_m + \frac{\gamma C_m}{1 - \gamma} \right). \tag{39}$$

### A.5 PROOF OF THEOREM 3 – EPISODIC ATTACK COST BOUND

According to the Corollary 2 in CPO (Achiam et al., 2017),

$$V_c^{\pi \circ \nu}(\mu_0) - V_c^{\pi}(\mu_0) \le \frac{1}{1 - \gamma} \mathbb{E}_{s \sim d_{\pi}, a \sim \pi \circ \nu} \left[ A_c^{\pi}(s,a) + \frac{2\gamma \delta_c^{\pi \circ \nu}}{1 - \gamma} D_{\mathrm{TV}}[\pi'(\cdot|s)\|\pi(\cdot|s)] \right], \tag{40}$$

where $\delta_c^{\pi \circ \nu} = \max_s |\mathbb{E}_{a \sim \pi \circ \nu} A_c^{\pi}(s,a)|$ and $A_c^{\pi}(s,a) = \mathbb{E}_{s' \sim p(\cdot|s,a)}[c(s,a,s') + \gamma V_c^{\pi}(s') - V_c^{\pi}(s)]$ denotes the advantage function. Note that

$$\mathbb{E}_{a \sim \pi \circ \nu} A_c^{\pi}(s,a) = \mathbb{E}_{a \sim \pi \circ \nu}[\mathbb{E}_{s' \sim p(\cdot|s,a)}[c(s,a,s') + \gamma V_c^{\pi}(s') - V_c^{\pi}(s)]] \tag{41}$$

$$= \mathbb{E}_{a \sim \pi \circ \nu, s' \sim p(\cdot|s,a)}[c(s,a,s') + \gamma V_c^{\pi}(s')] - V_c^{\pi}(s) \tag{42}$$

$$= \tilde{V}_c^{\pi, \nu}(s) - V_c^{\pi}(s). \tag{43}$$

By theorem 2,

$$\delta_c^{\pi \circ \nu} = \max_s |\mathbb{E}_{a \sim \pi \circ \nu} A_c^\pi(s, a)| \tag{44}$$

$$\leq \max_s \left| 2L\epsilon \left( p_s C_m + \frac{\gamma C_m}{1 - \gamma} \right) \right| \tag{45}$$

$$= 2L\epsilon C_m \left( \max_s p_s + \frac{\gamma}{1 - \gamma} \right). \tag{46}$$

Therefore, we can derive

$$V_c^{\pi \circ \nu}(\mu_0) - V_c^\pi(\mu_0) \leq \frac{1}{1 - \gamma} \max_s |\mathbb{E}_{a \sim \pi \circ \nu} A_c^\pi(s, a)| + \frac{2\gamma \delta_c^{\pi \circ \nu}}{(1 - \gamma)^2} D_{\text{TV}}[\pi'(\cdot|s) \| \pi(\cdot|s)] \tag{47}$$

$$= \left( \frac{1}{1 - \gamma} + \frac{2\gamma D_{\text{TV}}}{(1 - \gamma)^2} \right) \delta_c^{\pi \circ \nu} \tag{48}$$

$$\leq 2L\epsilon C_m \left( \frac{1}{1 - \gamma} + \frac{4\gamma L\epsilon}{(1 - \gamma)^2} \right) \left( \max_s p_s + \frac{\gamma}{1 - \gamma} \right). \tag{49}$$

Note $\pi$ is a feasible policy, i.e., $V_c^\pi(\mu_0) \leq \kappa$. Therefore,

$$V_c^{\pi \circ \nu}(\mu_0) \leq \kappa + 2L\epsilon C_m \left( \frac{1}{1 - \gamma} + \frac{4\gamma L\epsilon}{(1 - \gamma)^2} \right) \left( \max_s p_s + \frac{\gamma}{1 - \gamma} \right). \tag{50}$$

### A.6 PROOF OF THEOREM 4 AND PROPOSITION 1 – BELLMAN CONTRACTION

Recall Theorem 4, the Bellman policy operator $\mathcal{T}_\pi$ is a contraction under the sup-norm $\| \cdot \|_\infty$ and will converge to its fixed point. The Bellman policy operator is defined as:

$$(\mathcal{T}_\pi V_f^{\pi \circ \nu})(s) = \sum_{a \in \mathcal{A}} \pi(a|\nu(s)) \sum_{s' \in \mathcal{S}} p(s'|s, a) \left[ f(s, a, s') + \gamma V_f^{\pi \circ \nu}(s') \right], \quad f \in \{r, c\}, \tag{51}$$

The proof is as follows:

*Proof.* Denote $f_{sa}^{s'} = f(s, a, s'), f \in \{r, c\}$ and $p_{sa}^{s'} = p(s'|s, a)$ for simplicity, we have:

$$\left| (\mathcal{T}_\pi U_f^{\pi \circ \nu})(s) - (\mathcal{T}_\pi V_f^{\pi \circ \nu})(s) \right| = \left| \sum_{a \in \mathcal{A}} \pi(a|\nu(s)) \sum_{s' \in \mathcal{S}} p_{sa}^{s'} \left[ f_{sa}^{s'} + \gamma U_f^{\pi \circ \nu}(s') \right] \right. \tag{52}$$

$$\left. - \sum_{a \in \mathcal{A}} \pi(a|\nu(s)) \sum_{s' \in \mathcal{S}} p_{sa}^{s'} \left[ f_{sa}^{s'} + \gamma V_f^{\pi \circ \nu}(s') \right] \right| \tag{53}$$

$$= \gamma \left| \sum_{a \in \mathcal{A}} \pi(a|\nu(s)) \sum_{s' \in \mathcal{S}} p_{sa}^{s'} \left[ U_f^{\pi \circ \nu}(s') - V_f^{\pi \circ \nu}(s') \right] \right| \tag{54}$$

$$\leq \gamma \max_{s' \in \mathcal{S}} \left| U_f^{\pi \circ \nu}(s') - V_f^{\pi \circ \nu}(s') \right| \tag{55}$$

$$= \gamma \left\| U_f^{\pi \circ \nu}(s') - V_f^{\pi \circ \nu}(s') \right\|_\infty, \tag{56}$$

Since the above holds for any state $s$, we have:

$$\max_s \left| (\mathcal{T}_\pi U_f^{\pi \circ \nu})(s) - (\mathcal{T}_\pi V_f^{\pi \circ \nu})(s) \right| \leq \gamma \left\| U_f^{\pi \circ \nu}(s') - V_f^{\pi \circ \nu}(s') \right\|_\infty,$$

which implies that:

$$\left\| (\mathcal{T}_\pi U_f^{\pi \circ \nu})(s) - (\mathcal{T}_\pi V_f^{\pi \circ \nu})(s) \right\|_\infty \leq \gamma \left\| V_f^{\pi \circ \nu_2}(s') - V_f^{\pi \circ \nu_2}(s') \right\|_\infty,$$

Then based on the Contraction Mapping Theorem (Meir & Keeler, 1969), we know that $\mathcal{T}_\pi$ has a unique fixed point $V_f^*(s), f \in \{r, c\}$ such that $V_f^*(s) = (\mathcal{T}_\pi V_f^*)(s)$. $\square$

With the proof of Bellman contraction, we show that why we can perform adversarial training successfully under observational attacks. Since the Bellman operator is a contraction for both reward and cost under adversarial attacks, we can accurately evaluate the performance of the corrupted policy in the policy evaluation phase. This is a crucial and strong guarantee for the success of adversarial training, because we can not improve the policy without well-estimated values.

Propisition 1 states that suppose a trained policy $\pi'$ under the MC attacker satisfies: $V_c^{\pi' \circ \nu_{\text{MC}}}(\mu_0) \leq \kappa$, then $\pi' \circ \nu$ is guaranteed to be feasible with any $B_p^\epsilon$ bounded adversarial perturbations. Similarly, suppose a trained policy $\pi'$ under the MR attacker satisfies: $V_c^{\pi' \circ \nu_{\text{MR}}}(\mu_0) \leq \kappa$, then $\pi' \circ \nu$ is guaranteed to be non-tempting with any $B_p^\epsilon$ bounded adversarial perturbations. Before proving it, we first give the following definitions and lemmas.

**Definition 6.** Define the Bellman adversary effectiveness operator as $\mathcal{T}_c^* : \mathbb{R}^{|\mathcal{S}|} \to \mathbb{R}^{|\mathcal{S}|}$:

$$(\mathcal{T}_c^* V_c^{\pi \circ \nu})(s) = \max_{\tilde{s} \in B_p^\epsilon(s)} \sum_{a \in \mathcal{A}} \pi(a|\tilde{s}) \sum_{s' \in \mathcal{S}} p(s'|s,a) \left[ c(s,a,s') + \gamma V_c^{\pi \circ \nu}(s') \right]. \tag{57}$$

**Definition 7.** Define the Bellman adversary reward stealthiness operator as $\mathcal{T}_r^* : \mathbb{R}^{|\mathcal{S}|} \to \mathbb{R}^{|\mathcal{S}|}$:

$$(\mathcal{T}_r^* V_r^{\pi \circ \nu})(s) = \max_{\tilde{s} \in B_p^\epsilon(s)} \sum_{a \in \mathcal{A}} \pi(a|\tilde{s}) \sum_{s' \in \mathcal{S}} p(s'|s,a) \left[ r(s,a,s') + \gamma V_r^{\pi \circ \nu}(s') \right]. \tag{58}$$

Recall that $B_p^\epsilon(s)$ is the $\ell_p$ ball to constrain the perturbation range. The two definitions correspond to computing the value of the most effective and the most reward-stealthy attackers, which is similar to the Bellman optimality operator in the literature. We then show their contraction properties via the following Lemma:

**Lemma 3.** The Bellman operators $\mathcal{T}_c^*, \mathcal{T}_r^*$ are contractions under the sup-norm $\| \cdot \|_\infty$ and will converge to their fixed points, respectively. The fixed point for $\mathcal{T}_c^*$ is $V_c^{\pi \circ \nu_{\text{MC}}} = \mathcal{T}_c^* V_c^{\pi \circ \nu_{\text{MC}}}$, and the fixed point for $\mathcal{T}_r^*$ is $V_r^{\pi \circ \nu_{\text{MR}}} = \mathcal{T}_r^* V_r^{\pi \circ \nu_{\text{MR}}}$.

To finish the proof of Lemma 3, we introduce another lemma:

**Lemma 4.** Suppose $\max_x h(x) \geq \max_x g(x)$ and denote $x^{h*} = \arg\max_x h(x)$, we have:

$$\begin{aligned} |\max_x h(x) - \max_x g(x)| = \max_x h(x) - \max_x g(x) = h(x^{h*}) - \max_x g(x) \\ \leq h(x^{h*}) - g(x^{h*}) \leq \max_x |h(x) - g(x)|. \end{aligned} \tag{59}$$

We then prove the Bellman contraction properties of Lemma 3:

*Proof.*

$$\left| (\mathcal{T}_f^* V_f^{\pi \circ \nu_1})(s) - (\mathcal{T}_f^* V_f^{\pi \circ \nu_2})(s) \right| = \left| \max_{\tilde{s} \in B_p^\epsilon(s)} \sum_{a \in \mathcal{A}} \pi(a|\tilde{s}) \sum_{s' \in \mathcal{S}} p_{sa}^{s'} \left[ f_{sa}^{s'} + \gamma V_f^{\pi \circ \nu_1}(s') \right] \right. \tag{60}$$

$$\left. - \max_{\tilde{s} \in B_p^\epsilon(s)} \sum_{a \in \mathcal{A}} \pi(a|\tilde{s}) \sum_{s' \in \mathcal{S}} p_{sa}^{s'} \left[ f_{sa}^{s'} + \gamma V_f^{\pi \circ \nu_2}(s') \right] \right| \tag{61}$$

$$= \left| \gamma \max_{\tilde{s} \in B_p^\epsilon(s)} \sum_{a \in \mathcal{A}} \pi(a|\tilde{s}) \sum_{s' \in \mathcal{S}} p_{sa}^{s'} \left[ V_f^{\pi \circ \nu_1}(s') - V_f^{\pi \circ \nu_2}(s') \right] \right| \tag{62}$$

$$\leq \gamma \max_{\tilde{s} \in B_p^\epsilon(s)} \left| \sum_{a \in \mathcal{A}} \pi(a|\tilde{s}) \sum_{s' \in \mathcal{S}} p_{sa}^{s'} \left[ V_f^{\pi \circ \nu_1}(s') - V_f^{\pi \circ \nu_2}(s') \right] \right| \tag{63}$$

$$\triangleq \gamma \left| \sum_{a \in \mathcal{A}} \pi(a|\tilde{s}^*) \sum_{s' \in \mathcal{S}} p_{sa}^{s'} \left[ V_f^{\pi \circ \nu_1}(s') - V_f^{\pi \circ \nu_2}(s') \right] \right| \tag{64}$$

$$\leq \gamma \max_{s' \in \mathcal{S}} \left| V_f^{\pi \circ \nu_1}(s') - V_f^{\pi \circ \nu_2}(s') \right| \tag{65}$$

$$= \gamma \left\| V_f^{\pi \circ \nu_1}(s') - V_f^{\pi \circ \nu_2}(s') \right\|_\infty, \tag{66}$$

where inequality (63) comes from Lemma 4, and $\tilde{s}^*$ in Eq. (64) denote the argmax of the RHS.

Since the above holds for any state $s$, we can also conclude that:

$$\left\|(\mathcal{T}_f^* V_f^{\pi \circ \nu_1})(s) - (\mathcal{T}_f^* V_f^{\pi \circ \nu_2})(s)\right\|_\infty \leq \gamma \left\|V_f^{\pi \circ \nu_2}(s') - V_f^{\pi \circ \nu_2}(s')\right\|_\infty,$$

$\square$

After proving the contraction, we prove that the value function of the MC and MR adversaries $V_c^{\pi \circ \nu_{\text{MC}}}(s), V_r^{\pi \circ \nu_{\text{MR}}}(s)$ are the fixed points for $\mathcal{T}_c^*, \mathcal{T}_r^*$ as follows:

*Proof.* Recall that the MC, MR adversaries are:

$$\nu_{\text{MC}}(s) = \arg \max_{\tilde{s} \in B_p^\epsilon(s)} \mathbb{E}_{\tilde{a} \sim \pi(a|\tilde{s})} \left[Q_c^\pi(s, \tilde{a})\right], \nu_{\text{MR}}(s) = \arg \max_{\tilde{s} \in B_p^\epsilon(s)} \mathbb{E}_{\tilde{a} \sim \pi(a|\tilde{s})} \left[Q_r^\pi(s, \tilde{a})\right]. \quad (67)$$

Based on the value function definition, we have:

$$V_c^{\pi \circ \nu_{\text{MC}}}(s) = \mathbb{E}_{\tau \sim \pi \circ \nu_{\text{MC}}, s_0 = s}\left[\sum_{t=0}^\infty \gamma^t c_t\right] = \mathbb{E}_{\tau \sim \pi \circ \nu_{\text{MC}}, s_0 = s}\left[c_0 + \gamma \sum_{t=1}^\infty \gamma^{t-1} c_t\right] \quad (68)$$

$$= \sum_{a \in \mathcal{A}} \pi(a|\nu_{\text{MC}}(s)) \sum_{s' \in \mathcal{S}} p_{sa}^{s'} \left[c(s, a, s') + \gamma \mathbb{E}_{\tau \sim \pi \circ \nu_{\text{MC}}, s_1 = s'}\left[\sum_{t=1}^\infty \gamma^{t-1} c_t\right]\right] \quad (69)$$

$$= \sum_{a \in \mathcal{A}} \pi(a|\nu_{\text{MC}}(s)) \sum_{s' \in \mathcal{S}} p_{sa}^{s'} \left[c(s, a, s') + \gamma V_c^{\pi \circ \nu_{\text{MC}}}(s')\right] \quad (70)$$

$$= \max_{\tilde{s} \in B_p^\epsilon(s)} \sum_{a \in \mathcal{A}} \pi(a|\tilde{s}) \sum_{s' \in \mathcal{S}} p_{sa}^{s'} \left[c(s, a, s') + \gamma V_c^{\pi \circ \nu_{\text{MC}}}(s')\right] \quad (71)$$

$$= (\mathcal{T}_c^* V_c^{\pi \circ \nu_{\text{MC}}})(s), \quad (72)$$

where Eq. (71) is from the MC attacker definition. Therefore, the cost value function of the MC attacker $V_c^{\pi \circ \nu_{\text{MC}}}$ is the fixed point of the Bellman adversary effectiveness operator $\mathcal{T}_c^*$. With the same procedure (replacing $\nu_{\text{MC}}, \mathcal{T}_c^*$ with $\nu_{\text{MR}}, \mathcal{T}_r^*$), we can prove that the reward value function of the MR attacker $V_r^{\pi \circ \nu_{\text{MR}}}$ is the fixed point of the Bellman adversary stealthiness operator $\mathcal{T}_r^*$.

$\square$

With Lemma 3 and the proof above, we can easily obtain the conclusions in Remark 1: if the trained policy is safe under the MC or the MR attacker, then it is guaranteed to be feasible or non-tempting under any $B_p^\epsilon(s)$ bounded adversarial perturbations respectively, since there are no other attackers can achieve higher cost or reward returns than them. It provides theoretical guarantees of the safety of adversarial training under the MC and MR attackers. The adversarial trained agents under the proposed attacks are guaranteed to be safe or non-tempting under any bounded adversarial perturbations. We believe the above theoretical guarantees are crucial for the success of our adversarial training agents, because from our ablation studies, we can see adversarial training can not achieve desired performance with other attackers.

## B REMARKS

### B.1 REMARKS OF THE SAFE RL SETTING, STEALTHINESS, AND ASSUMPTIONS

**Safe RL setting regarding the reward and the cost.** We consider the safe RL problems that have separate task rewards and constraint violation costs, i.e. independent reward and cost functions. Combining the cost with reward to a single scalar metric, which can be viewed as manually selecting Lagrange multipliers, may work in simple problems. However, it lacks interpretability – it is hard to explain what does a single scalar value mean, and requires good domain knowledge of the problem – the weight between costs and rewards should be carefully balanced, which is difficult when the task rewards already contain many objectives/factors. On the other hand, separating the costs from rewards is easy to monitor the safety performance and task performance respectively, which is more interpretable and applicable for different cost constraint thresholds.

**Determine temptation status of a safe RL problem.** According to Def. 1-3, no tempting policy indicates a non-tempting safe RL problem, where the optimal policy has the highest reward while satisfying the constraint. However, for the safe deployment problem that only cares about safety after training, no tempting policy means that the cost signal is unnecessary for training, because one can simply focus on maximizing the reward. As long as the most rewarding policies are found, the safety requirement would be automatically satisfied, and thus many standard RL algorithms can solve the problem. Since safe RL methods are not required in this setting, the non-tempting tasks are usually not discussed in safe RL papers, and are also not the focus of this paper. From another perspective, since a safe RL problem is specified by the cost threshold $\kappa$, one can tune the threshold to change the status of temptation. For instance, if $\kappa > \max_{s,a,s'} c(s, a, s')$, then it is guaranteed to be a non-tempting problem because all the policies satisfy the constraints, and thus we can use standard RL methods to solve it.

**Independently estimated reward and cost value functions assumption.** Similar to most existing safe RL algorithms, such as PPO-Lagrangian Ray et al. (2019); Stooke et al. (2020), CPO Achiam et al. (2017), FOCOPS Zhang et al. (2020b), and CVPO Liu et al. (2022), we consider the policy-based (or actor-critic-based) safe RL in this work. There are two phases for this type of approach: policy evaluation and policy improvement. In the **policy evaluation** phase, the reward and cost value functions $V_r^\pi, V_c^\pi$ are evaluated separately. At this stage, the Bellman operators for reward and cost values are *independent*. Therefore, they have contractions (Theorem 4) and will converge to their fixed points separately. This is a commonly used treatment in safe RL papers to train the policy: first evaluating the reward and cost values independently by Bellman equations and then optimizing the policy based on the learned value estimations. Therefore, our theoretical analysis of robustness is also developed under this setting.

**(Reward) Stealthy attack for safe RL.** As we discussed in Sec. 3.2, the stealthiness concept in supervised learning refers to that the adversarial attack should be covert to prevent from being easily identified. While we use the perturbation set $B_p^\epsilon$ to ensure the stealthiness regarding the observation corruption, we notice that another level of stealthiness regarding the task reward performance is interesting and worthy of being discussed. In some real-world applications, the task-related metrics (such as velocity, acceleration, goal distances) are usually easy to be monitored from sensors. However, the safety metrics can be sparse and hard to monitor until breaking the constraints, such as colliding with obstacles and entering hazard states, which are determined by binary indicator signals. Therefore, a dramatic task-related metrics (reward) drop might be easily detected by the agent, while constraint violation signals could be hard to detect until catastrophic failures. An unstealthy attack in this scenario may decrease the reward a lot and prohibit the agent from finishing the task, which can warn the agent that it is attacked and thus lead to a failing attack. On the contrary, a stealthy attack can maintain the agent's task reward such that the agent is not aware of the existence of the attacks based on "good" task metrics, while performing successful attacks by leading to constraint violations. In other words, a stealthy attack should corrupt the policy to be tempted, since all the tempting policies are high-rewarding while unsafe.

**Stealthiness definition of the attacks.** There is an alternative definition of stealthiness by viewing the difference in the reward regardless of increasing or decreasing. The two-sided stealthiness is a more strict one than the one-sided lower-bound definition in this paper. However, if we consider a practical system design, people usually set a threshold for the lower bound of the task performance to determine whether the system functions properly, rather than specifying an upper bound of the performance because it might be tricky to determine what should be the upper-bound of the task performance to be alerted by the agent. For instance, an autonomous vehicle that fails to reach the destination within a certain amount of time may be identified as abnormal, while reaching the goal faster may not since it might be hard to specify such a threshold to determine what is an overly good performance. Therefore, increasing the reward with the same amount of decreasing it may not attract the same attention from the agents. In addition, finding a stealthy and effective attacker with minimum reward change might be a much harder problem with the two-sided definition, since the candidate solutions are much fewer and the optimization problem could be harder to be formulated. But we believe that this is an interesting point that is worthy to be investigated in the future, while we will focus on the one-sided definition of stealthiness in this work.

### B.2 REMARKS OF THE FAILURE OF SA-PPOL(MC/MR) BASELINES

The detailed algorithm of SA-PPOL Zhang et al. (2020a) can be found in Appendix C.5. The basic idea can be summarized via the following equation:

$$\ell_\nu(s) = -D_{KL}[\pi(\cdot|s)||\pi_\theta(\cdot|\nu(s))], \tag{73}$$

which aims to minimize the divergence between the corrupted states and the original states. Note that we only optimize (compute gradient) for $\pi_\theta(\cdot|\nu(s))$ rather than $\pi(\cdot|s)$, since we view $\pi(\cdot|s)$ as the "ground-truth" target action distribution. Adding the above KL regularizer to the original PPOL loss yields the SA-PPOL algorithm. We could observe the original SA-PPOL that uses the MAD attacker as the adversary can learn well in most of the tasks, though it is not safe under strong attacks. However, SA-PPOL with MR or MC adversaries often fail to learn a meaningful policy in many tasks, especially for the MR attacker. The reason is that: the MR attacker aims to find the high-rewarding adversarial states, while the KL loss will make the policy distribution of high-rewarding adversarial states to match with the policy distribution of the original relatively lower-rewards states. As a result, the training could fail due to wrong policy optimization direction and prohibited exploration to high-rewarding states. Since the MC attacker can also lead to high-rewarding adversarial states due to the existence of tempting polices, we may also observe failure training with the MC attacker.

## C IMPLEMENTATION DETAILS

### C.1 MC AND MR ATTACKERS IMPLEMENTATION

We use the gradient of the state-action value function $Q(s, a)$ to provide the direction to update states adversarially in K steps ($Q = Q_r^\pi$ for MR and $Q = Q_c^\pi$ for MC):

$$s^{k+1} = \text{Proj}[s^k - \eta\nabla_{s^k}Q(s^0, \pi(s^k))], k = 0, \ldots, K-1 \tag{74}$$

where $\text{Proj}[\cdot]$ is a projection to $B_p^\epsilon(s^0)$, $\eta$ is the learning rate, and $s^0$ is the state under attack. Since the Q-value function and policy are parametrized by neural networks, we can backpropagate the gradient from $Q_c$ or $Q_r$ to $s^k$ via $\pi(\tilde{a}|s^k)$, which can be solved efficiently by many optimizers like ADAM. It is related to the Projected Gradient Descent (PGD) attack, and the deterministic policy gradient method such as DDPG and TD3 in the literature, but the optimization variables are the state perturbations rather than the policy parameters.

Note that we use the gradient of $Q(s^0, \pi(s^k))$ rather than $Q(s^k, \pi(s^k))$ to make the optimization more stable, since the $Q$ function may not generalize well to unseen states in practice. This technique for solving adversarial attacks is also widely used in the standard RL literature and is shown to be successful, such as (Zhang et al., 2020a). The implementation of MC and MR attacker is shown in algorithm 2. Empirically, this gradient-based method converges fast with a few iterations and within 10ms as shown in Fig. 3, which greatly improves adversarial training efficiency.

---

**Algorithm 2** MC and MR attacker

---

**Input:** A policy $\pi$ under attack, corresponding $Q$ networks, initial state $s^0$, attack steps $K$, attacker learning rate $\eta$, perturbation range $\epsilon$, two thresholds $\epsilon_Q$ and $\epsilon_s$ for early stopping
**Output:** An adversarial state $\tilde{s}$

1: **for** $k = 1$ to $K$ **do**
2: $\quad g^k = \nabla_{s^{k-1}}Q(s_0, \pi(s^{k-1}))$
3: $\quad s^k \leftarrow \text{Proj}[s^{k-1} - \eta g^k]$
4: $\quad$ Compute $\delta Q = |Q(s_0, \pi(s^k)) - Q(s_0, \pi(s^{k-1}))|$ and $\delta s = |s^k - s^{k-1}|$
5: $\quad$ **if** $\delta Q < \epsilon_Q$ and $\delta s < \epsilon_s$ **then**
6: $\quad\quad$ break for early stopping
7: $\quad$ **end if**
8: **end for**

---

### C.2 PPO-LAGRANGIAN ALGORITHM

The objective of PPO (clipped) has the form (Schulman et al., 2017):

$$\ell_{ppo} = \min(\frac{\pi_\theta(a|s)}{\pi_{\theta_k}(a|s)}A^{\pi_{\theta_k}}(s, a), \text{clip}(\frac{\pi_\theta(a|s)}{\pi_{\theta_k}(a|s)}, 1-\epsilon, 1+\epsilon)A^{\pi_{\theta_k}}(s, a)) \tag{75}$$

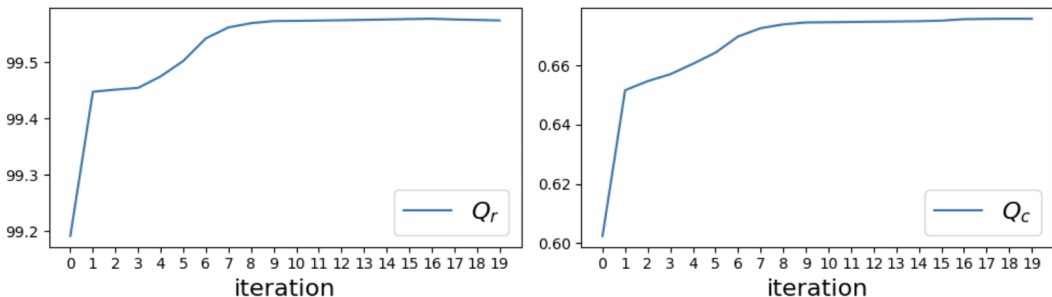

Figure 3: An example of updating the MR and MC during training.

We use PID Lagrangian Stooke et al. (2020) that addresses the oscillation and overshoot problem in Lagrangian methods. The loss of the PPO-Lagrangian has the form:

$$\ell_{ppol} = \frac{1}{1+\lambda}(\ell_{ppo} + V_r^\pi - \lambda V_c^\pi) \tag{76}$$

The Lagrangian multiplier $\lambda$ is computed by applying feedback control to $V_c^\pi$ and is determined by $K_P$, $K_I$, and $K_D$ that need to be fine-tuned.

### C.3 ADVERSARIAL TRAINING FULL ALGORITHM

Due to the page limit, we omit some implementation details in the main content. We will present the full algorithm and some implementation tricks in this section. Without otherwise statement, the critics' and policies' parameterization is assumed to be neural networks (NN), while we believe other parameterization form should also work well.

**Critics update**. Denote $\phi_r$ as the parameters for the task reward critic $Q_r$, and $\phi_c$ as the parameters for the constraint violation cost critic $Q_c$. Similar to many other off-policy algorithms Lillicrap et al. (2015), we use a target network for each critic and the polyak smoothing trick to stabilize the training. Other off-policy critics training methods, such as Re-trace Munos et al. (2016), could also be easily incorporated with PPO-Lagrangian training framework. Denote $\phi'_r$ as the parameters for the **target** reward critic $Q'_r$, and $\phi'_c$ as the parameters for the **target** cost critic $Q'_c$. Define $\mathcal{D}$ as the replay buffer and $(s, a, s', r, c)$ as the state, action, next state, reward, and cost respectively. The critics are updated by minimizing the following mean-squared Bellman error (MSBE):

$$\ell(\phi_r) = \mathbb{E}_{(s,a,s',r,c)\sim\mathcal{D}}\Big[ \big(Q_r(s,a) - (r + \gamma\mathbb{E}_{a'\sim\pi}[Q'_r(s',a')])\big)^2 \Big] \tag{77}$$

$$\ell(\phi_c) = \mathbb{E}_{(s,a,s',r,c)\sim\mathcal{D}}\Big[ \big(Q_c(s,a) - (c + \gamma\mathbb{E}_{a'\sim\pi}[Q'_c(s',a')])\big)^2 \Big]. \tag{78}$$

Denote $\alpha_c$ as the critics' learning rate, we have the following updating equations:

$$\phi_r \leftarrow \phi_r - \alpha_c\nabla_{\phi_r}\ell(\phi_r) \tag{79}$$
$$\phi_c \leftarrow \phi_c - \alpha_c\nabla_{\phi_c}\ell(\phi_c) \tag{80}$$

Note that the original PPO-Lagrangian algorithm is an on-policy algorithm, which doesn't require the reward critic and cost critic to train the policy. We learn the critics because the MC and MR attackers require them, which is an essential module for adversarial training.

**Polyak averaging for the target networks**. The polyak averaging is specified by a weight parameter $\rho \in (0, 1)$ and updates the parameters with:

$$\phi'_r = \rho\phi'_r + (1 - \rho)\phi_r$$
$$\phi'_c = \rho\phi'_c + (1 - \rho)\phi_c \tag{81}$$
$$\theta' = \rho\theta' + (1 - \rho)\theta.$$

The critic's training tricks are widely adopted in many off-policy RL algorithms, such as SAC, DDPG and TD3. We observe that the critics trained with those implementation tricks work well in practice. Then we present the full Robust PPO-Lagrangian algorithm:

---

**Algorithm 3** Robust PPO-Lagrangian Algorithm

---

**Input:** rollouts $T$, policy optimization steps $M$, PPO-Lag loss function $\ell_{ppol}(s, \pi_\theta, r, c)$, adversary function $\nu(s)$, policy parameter $\theta$, critic parameter $\phi_r$ and $\phi_c$, target critic parameter $\phi'_r$ and $\phi'_c$
**Output:** policy $\pi_\theta$
 1: Initialize policy parameters and critics parameters
 2: **for** each training iteration **do**
 3:    Rollout $T$ trajectories by $\pi_\theta \circ \nu$ from the environment $\{(\nu(s), \nu(a), \nu(s'), r, c)\}_N$
 4:    ▷ *Update learner*
 5:    **for** Optimization steps $m = 1, ..., M$ **do**
 6:        ▷ *No KL regularizer!*
 7:        Compute PPO-Lag loss $\ell_{ppol}(\tilde{s}, \pi_\theta, r, c)$ by Eq. (76)
 8:        Update actor $\theta \leftarrow \theta - \alpha \nabla_\theta \ell_{ppo}$
 9:    **end for**
10:    Update value function based on samples $\{(s, a, s', r, c)\}_N$
11:    ▷ *Update adversary scheduler*
12:    Update critics $Q_c$ and $Q_r$ by Eq. (79) and Eq. (80)
13:    Polyak averaging target networks by Eq. (81)
14:    Update current perturbation range
15:    Update adversary based on $Q_c$ and $Q_r$ using algorithm 2
16:    Linearly increase the perturbation range until to the maximum number $\epsilon$
17: **end for**

---

## C.4 MAD ATTACKER IMPLEMENTATION

The full algorithm of MAD attacker is presented in algorithm 4. We use the same SGLD optimizer as in Zhang et al. (2020a) to maximize the KL-divergence. The objective of the MAD attacker is defined as:

$$\ell_{MAD}(s) = -D_{KL}[\pi(\cdot|s_0)||\pi_\theta(\cdot|s)] \tag{82}$$

Note that we back-propagate the gradient from the corrupted state $s$ instead of the original state $s_0$ to the policy parameters $\theta$. The full algorithm is shown below:

---

**Algorithm 4** MAD attacker

---

**Input:** A policy $\pi$ under attack, corresponding $Q(s, a)$ network, initial state $s^0$, attack steps $K$, attacker learning rate $\eta$, the (inverse) temperature parameter for SGLD $\beta$, two thresholds $\epsilon_Q$ and $\epsilon_s$ for early stopping
**Output:** An adversarial state $\tilde{s}$
 1: **for** $k = 1$ to $K$ **do**
 2:    Sample $\upsilon \sim \mathcal{N}(0, 1)$
 3:    $g^k = \nabla \ell_{MAD}(s_{t-1}) + \sqrt{\frac{2}{\beta\eta}}\upsilon$
 4:    $s^k \leftarrow \text{Proj}[s^{k-1} - \eta g^k]$
 5:    Compute $\delta Q = |Q(s_0, \pi(s^k)) - Q(s_0, \pi(s^{k-1}))|$ and $\delta s = |s^k - s^{k-1}|$
 6:    **if** $\delta Q < \epsilon_Q$ and $\delta s < \epsilon_s$ **then**
 7:        break for early stopping
 8:    **end if**
 9: **end for**

---

## C.5 SA-PPO-LAGRANGIAN BASELINE

---

**Algorithm 5** SA-PPO-Lagrangian Algorithm

---

**Input:** rollouts $T$, policy optimization steps $M$, PPO-Lag loss function $\ell_{ppo}(s, \pi_\theta, r, c)$, adversary function $\nu(s)$

**Output:** policy $\pi_\theta$

1: Initialize policy parameters and critics parameters
2: **for** each training iteration **do**
3:     Rollout $T$ trajectories by $\pi_\theta$ from the environment $\{(s, a, s', r, c)\}_N$
4:     Compute adversary states $\tilde{s} = \nu(s)$ for the sampled trajectories
5:     ▷ *Update actors*
6:     **for** Optimization steps $m = 1, ..., M$ **do**
7:         Compute KL robustness regularizer $\tilde{L}_{KL} = D_{\mathrm{KL}}(\pi(s)\|\pi_\theta(\tilde{s}))$, no gradient from $\pi(s)$
8:         Compute PPO-Lag loss $\ell_{ppol}(s, \pi_\theta, r, c)$ by Eq. (76)
9:         Combine them together with a weight $\beta$: $\ell = \ell_{ppol}(s, \pi_\theta, r, c) + \beta\tilde{\ell}_{KL}$
10:        Update actor $\theta \leftarrow \theta - \alpha\nabla_\theta\ell$
11:     **end for**
12:     ▷ *Update critics*
13:     Update value function based on samples $\{(s, a, s', r, c)\}_N$
14: **end for**

---

The SA-PPO-Lagrangian algorithm adds an additional KL robustness regularizer to robustify the training policy. Choosing different adversaries $\nu$ yields different baseline algorithms. The original SA-PPOL (Zhang et al., 2020a) method adopts the MAD attacker, while we conduct ablation studies by using the MR attacker and the MC attacker, which yields the SA-PPOL(MR) and the SA-PPOL(MC) baselines respectively.

## C.6 IMPROVED ADAPTIVE MAD (AMAD) ATTACKER BASELINE

To motivate the design of AMAD baseline, we denote $P^\pi(s'|s) = \int p(s'|s, a)\pi(a|s)da$ as the state transition kernel and $p_t^\pi(s) = p(s_t = s|\pi)$ as the probability of visiting the state $s$ at the time $t$ under the policy $\pi$, where $p_t^\pi(s') = \int P^\pi(s'|s)p_{t-1}^\pi(s)ds$. Then the discounted future state distribution $d^\pi(s)$ is defined as (Kakade, 2003):

$$d^\pi(s) = (1 - \gamma)\sum_{t=0}^{\infty}\gamma^t p_t^\pi(s),$$

which allows us to represent the value functions compactly:

$$
\begin{aligned}
V_f^\pi(\mu_0) &= \frac{1}{1 - \gamma}\mathbb{E}_{s\sim d^\pi, a\sim\pi, s'\sim p}[f(s, a, s')] \\
&= \frac{1}{1 - \gamma}\int_{s\in\mathcal{S}}d^\pi(s)\int_{a\in\mathcal{A}}\pi(a|s)\int_{s'\in\mathcal{S}}p(s'|s, a)f(s, a, s')ds'dads, \quad f \in \{r, c\}
\end{aligned}
\tag{83}
$$

Based on Lemma 2, the optimal policy $\pi^*$ in a tempting safe RL setting satisfies:

$$\frac{1}{1 - \gamma}\int_{s\in\mathcal{S}}d^{\pi^*}(s)\int_{a\in\mathcal{A}}\pi^*(a|s)\int_{s'\in\mathcal{S}}p(s'|s, a)c(s, a, s')ds'dads = \kappa. \tag{84}$$

We can see that performing MAD attack in low-risk regions that with small $p(s'|s, a)c(s, a, s')$ values may not be effective – the agent may not even be close to the safety boundary. On the other hand, perturbing $\pi$ when $p(s'|s, a)c(s, a, s')$ is large may have higher chance to result in constraint violations. Therefore, we improve the MAD to the Adaptive MAD attacker, which will only attack the agent in high-risk regions (determined by the cost value function and a threshold $\xi$).

The implementation of AMAD is shown in algorithm 6. Given a batch of states $\{s\}_N$, we compute the cost values $\{V_c^\pi(s)\}_N$ and sort them in ascending order. Then we select certain percentile of $\{V_c^\pi(s)\}_N$ as the threshold $\xi$ and attack the states that have higher cost value than $\xi$.

---

**Algorithm 6** AMAD attacker

---

**Input:** a batch of states $\{s\}_N$, threshold $\xi$, a policy $\pi$ under attack, corresponding $Q(s, a)$ network, initial state $s^0$, attack steps $K$, attacker learning rate $\eta$, the (inverse) temperature parameter for SGLD $\beta$, two thresholds $\epsilon_Q$ and $\epsilon_s$ for early stopping
**Output:** batch adversarial state $\tilde{s}$

1: Compute batch cost values $\{V_c^\pi(s)\}_N$
2: $\xi \leftarrow (1 - \xi)$ percentile of $V_c^\pi(s)$
3: **for** the state $s$ that $V_c^\pi(s) > \xi$ **do**
4:     compute adversarial state $\tilde{s}$ by algorithm 4
5: **end for**

---

## C.7 ENVIRONMENT DESCRIPTION

We use the Bullet safety gym (Gronauer, 2022) environments for this set of experiments. In the Circle tasks, the goal is for an agent to move along the circumference of a circle while remaining within a safety region smaller than the radius of the circle. The reward and cost functions are defined as:

$$r(s) = \frac{-yv_x + xv_y}{1 + |\sqrt{x^2 + y^2} - r|} + r_{robot}(s)$$
$$c(s) = \mathbf{1}(|x| > x_{lim})$$

where $x, y$ are the position of the agent on the plane, $v_x, v_y$ are the velocities of the agent along the $x$ and $y$ directions, $r$ is the radius of the circle, and $x_{lim}$ specified the range of the safety region, $r_{robot}(s)$ is the specific reward for different robot. For example, an ant robot will gain reward if its feet do not collide with each other. In the Run tasks, the goal for an agent is to move as far as possible within the safety region and the speed limit. The reward and cost functions are defined as:

$$r(s) = \sqrt{(x_{t-1} - g_x)^2 + (y_{t-1} - g_y)^2} - \sqrt{(x_t - g_x)^2 + (y_t - g_y)^2} + r_{robot}(s)$$
$$c(s) = \mathbf{1}(|y| > y_{lim}) + \mathbf{1}(\sqrt{v_x^2 + v_y^2} > v_{lim})$$

where $v_{lim}$ is the speed limit and $g_x$ and $g_y$ is the position of a fictitious target. The reward is the difference between current distance to the target and the distance in the last timestamp.

## C.8 HYPER-PARAMETERS

In all experiments, we use Gaussian policies with mean vectors given as the outputs of neural networks, and with variances that are separate learnable parameters. For the Car-Run experiment, the policy networks and Q networks consist of two hidden layers with sizes of (128, 128). For other experiments, they have two hidden layers with sizes of (256, 256). In both cases, the ReLU activation function is used. We use a discount factor of $\gamma = 0.995$, a GAE-$\lambda$ for estimating the regular advantages of $\lambda^{GAE} = 0.97$, a KL-divergence step size of $\delta_{KL} = 0.01$, a clipping coefficient of 0.02. The PID parameters for the Lagrange multiplier are: $K_p = 0.1$, $K_I = 0.003$, and $K_D = 0.001$. The learning rate of the adversarial attackers: MAD, AMAD, MC, and MR is 0.05. The optimization steps of MAD and AMAD is 60 and 200 for MC and MR attacker. The threshold $\xi$ for AMAD is 0.1. The complete hyperparameters used in the experiments are shown in Table 2. We choose larger perturbation range for the Car robot-related tasks because they are simpler and easier to train.

Table 2: Hyperparameters for all the environments

| Parameter | Car-Run | Drone-Run | Ant-Run | Car-Circle | Drone-Circle | Ant-Circle |
|---|---|---|---|---|---|---|
| training epoch | 100 | 250 | 250 | 100 | 500 | 800 |
| batch size | 40000 | 80000 | 80000 | 40000 | 60000 | 80000 |
| minibatch size | 300 | 300 | 300 | 300 | 300 | 300 |
| rollout length | 200 | 100 | 200 | 300 | 300 | 300 |
| cost limit | 5 | 5 | 5 | 5 | 5 | 5 |
| perturbation $\epsilon$ | 0.05 | 0.025 | 0.025 | 0.05 | 0.025 | 0.025 |
| actor optimization step $M$ | 80 | 80 | 80 | 80 | 80 | 160 |
| actor learning rate | 0.0003 | 0.0002 | 0.0005 | 0.0003 | 0.0003 | 0.0005 |
| critic learning rate | 0.001 | 0.001 | 0.001 | 0.001 | 0.001 | 0.001 |

## C.9 More Experiment Results

All the experiments are performed on a server with AMD EPYC 7713 64-Core Processor CPU. For each experiment, we use 4 CPUs to train each agent that is implemented by PyTorch, and the training time varies from 4 hours (Car-Run) to 7 days (Ant-Circle). Video demos are available at: https://sites.google.com/view/robustsaferl/home

The experiments for the minimizing reward attack for our method are shown in Table 3. We can see that the minimizing reward attack does not have an effect on the cost since it remains below the constraint violation threshold. Besides, we adopted one SOTA attack method (MAD) in standard RL as a baseline, and improve it (AMAD) in the safe RL setting. The results, however, demonstrate that they do not perform well. As a result, it does not necessarily mean that the attacking methods and robust training methods in standard RL settings still perform well in the safe RL setting.

Table 3: Evaluation results under Minimum Reward attacker. Each value is reported as: mean and the difference between the natural performance for 50 episodes and 5 seeds.

| Method | | Car-Run $\epsilon = 0.05$ | Drone-Run $\epsilon = 0.025$ | Ant-Run $\epsilon = 0.025$ | Ant-Circle $\epsilon = 0.025$ |
|---|---|---|---|---|---|
| PPOL-vanilla | Reward | 496.65 ($\downarrow$64.68) | 265.06($\downarrow$82.11) | 498.42($\downarrow$179.98) | 67.9($\downarrow$89.54) |
| | Cost | 0.0($\downarrow$0.15) | 0.0($\downarrow$0.0) | 0.03($\downarrow$1.2) | 1.17($\downarrow$1.53) |
| ADV-PPOL(MC) | Reward | 491.95($\downarrow$33.81) | 211.16($\downarrow$62.24) | 548.0($\downarrow$53.25) | 86.26($\downarrow$49.72) |
| | Cost | 0.0($\downarrow$0.0) | 0.4($\uparrow$0.4) | 0.0($\downarrow$0.0) | 0.0($\downarrow$0.3) |
| ADV-PPOL(MR) | Reward | 491.48($\downarrow$34.45) | 214.25($\downarrow$19.06) | 524.24($\downarrow$95.93) | 87.24($\downarrow$46.03) |
| | Cost | 0.0($\downarrow$0.0) | 1.1($\uparrow$1.1) | 0.0($\downarrow$0.17) | 1.4($\uparrow$0.53) |

The experiment results of FOCOPS (Zhang et al., 2020b) is shown in Table 4. We trained FO-COPS without adversarial attackers FOCOPS-vanilla and with our adversarial training methods FOCOPS(MR) and FOCOPS(MR) under the MC and MR attackers respectively. We can see that the vanilla method is safe in noise-free environments, however, they are not safe anymore under the proposed adversarial attack. In addition, the adversarial training can help to improve the robustness and make the FOCOPS agents much safer under strong attacks, which means that our adversarial training method is generalizable to different safe RL methods.

Table 4: Evaluation results of natural performance (no attack) and under MAD, MC, and MR attackers of FOCOPS. Each value is reported as: mean ± standard deviation for 50 episodes and 5 seeds.

| Env | Method | Natural | | MAD | | MC | | MR | |
|---|---|---|---|---|---|---|---|---|---|
| | | Reward | Cost | Reward | Cost | Reward | Cost | Reward | Cost |
| Car-Circle $\epsilon = 0.05$ | FOCOPS-vanilla | 304.2±16.91 | 0.0±0.0 | 307.08±42.04 | 19.94±16.05 | 286.66±53.7 | 31.25±18.08 | 382.99±22.86 | 48.88±14.25 |
| | FOCOPS(MC) | 268.56±44.79 | 0.0±0.0 | 256.05±45.26 | 0.0±0.0 | 284.93±45.84 | 0.97±2.99 | 267.37±49.75 | 0.64±1.92 |
| | FOCOPS(MR) | 305.91±18.16 | 0.0±0.0 | 295.86±20.02 | 0.04±0.4 | 264.33±25.76 | 1.64±3.62 | 308.62±26.33 | 0.82±1.98 |
| Car-Run $\epsilon = 0.05$ | FOCOPS-vanilla | 509.47±11.7 | 0.0±0.0 | 494.74±11.75 | 0.95±1.32 | 540.23±12.56 | 27.0±17.61 | 539.85±11.85 | 25.1±17.29 |
| | FOCOPS(MC) | 473.47±5.89 | 0.0±0.0 | 460.79±7.76 | 0.0±0.0 | 495.54±9.83 | 0.45±1.15 | 497.24±6.6 | 0.62±1.23 |
| | FOCOPS(MR) | 486.98±5.53 | 0.0±0.0 | 434.96±19.79 | 0.0±0.0 | 488.24±23.98 | 0.62±1.1 | 488.58±24.65 | 0.52±0.9 |

We evaluate the performance of MAD and AMAD adversaries by attacking well-trained PPO-Lagrangian policies. We keep the policies' model weights fixed for all the attackers. The comparison is in Fig. 4. We vary the attacking fraction (determined by $\xi$) to thoroughly study the effectiveness of the AMAD attacker. We can see that AMAD attacker is more effective because the cost increases significantly with the increase in perturbation, while the reward is maintained well. This validates our hypothesis that attacking the agent in high-risk regions is more effective and stealthy.

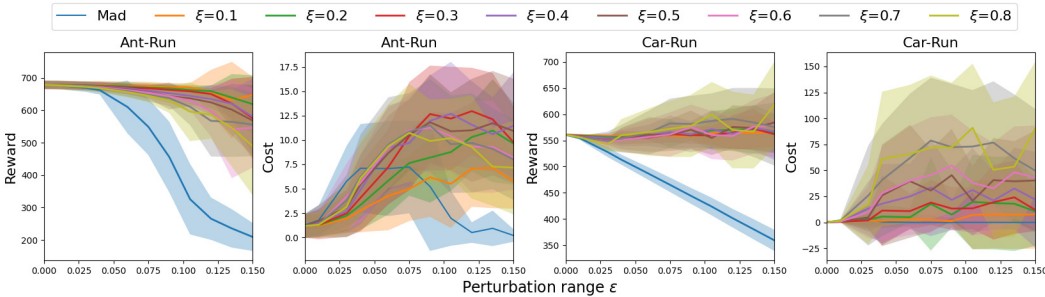

Figure 4: Reward and cost of AMAD and MAD attacker

The experiment results of trained safe RL policies under the Random and MAD attackers are shown in Table 5. The last column shows the average rewards and costs over all the 5 attackers (Random, MAD, AMAD, MC, MR). Our agent (ADV-PPOL) with adversarial training is robust against all the 5 attackers and achieves the lowest cost. We can also see that AMAD attacker is more effective than MAD since the cost under the AMAD attacker is higher than the cost under the MAD attacker.

Table 5: Evaluation results of natural performance (no attack) and under Random and MAD attackers. The average column shows the average rewards and costs over all 5 attackers (Random, MAD, AMAD, MC, and MR). Our methods are ADV-PPOL(MC/MR). Each value is reported as: mean ± standard deviation for 50 episodes and 5 seeds. We shadow two lowest-costs agents under each attacker column and break ties based on rewards, excluding the failing agents (whose natural rewards are less than 30% of PPOL-vanilla's). We mark the failing agents with ⋆.

| Env | Method | Random | | MAD | | Average | |
|---|---|---|---|---|---|---|---|
| | | Reward | Cost | Reward | Cost | Reward | Cost |
| Car-Run ε = 0.05 | PPOL-vanilla | 552.54±2.05 | 16.66±6.16 | 500.72±15.91 | 0.04±0.57 | 570.6±2.5 | 64.78±1.55 |
| | PPOL-random | 555.5±1.46 | 1.0±1.4 | 543.64±2.4 | 3.52±2.92 | 563.3±1.03 | 62.08±0.68 |
| | SA-PPOL | 533.74±9.02 | 0.0±0.0 | 533.84±10.79 | 0.0±0.0 | 544.9±8.48 | 4.26±3.89 |
| | SA-PPOL(MC) | 539.57±3.53 | 0.0±0.0 | 520.11±4.3 | 0.0±0.0 | 549.93±3.18 | 1.0±1.7 |
| | SA-PPOL(MR) | 541.63±2.21 | 0.0±0.0 | 528.32±4.55 | 0.0±0.06 | 550.58±2.72 | 8.34±16.69 |
| | ADV-PPOL(MC) | 500.99±9.9 | 0.0±0.0 | 464.06±17.13 | 0.0±0.0 | 512.72±7.42 | 0.0±0.03 |
| | ADV-PPOL(MR) | 492.01±8.15 | 0.0±0.0 | 449.23±13.21 | 0.0±0.0 | 505.79±5.46 | 0.01±0.04 |
| Drone-Run ε = 0.025 | PPOL-vanilla | 346.67±2.4 | 20.53±8.56 | 350.92±38.17 | 48.3±30.25 | 347.88±6.64 | 38.07±5.5 |
| | PPOL-random | 341.5±1.51 | 1.22±2.44 | 274.22±24.53 | 5.62±19.94 | 316.44±5.71 | 11.92±6.1 |
| | SA-PPOL | 335.7±7.01 | 11.09±13.08 | 370.3±47.74 | 56.94±24.11 | 330.18±39.47 | 32.76±6.94 |
| | SA-PPOL(MC) | 183.11±19.96 | 0.0±0.0 | 65.37±34.6 | 0.0±0.0 | 199.43±17.87 | 7.69±5.13 |
| | *SA-PPOL(MR) | 0.42±0.72 | 0.0±0.0 | 0.86±1.6 | 0.0±0.0 | 0.42±0.72 | 0.0±0.0 |
| | ADV-PPOL(MC) | 260.51±9.7 | 0.0±0.0 | 254.41±29.96 | 0.0±0.0 | 266.92±16.5 | 2.75±2.79 |
| | ADV-PPOL(MR) | 223.89±75.94 | 0.0±0.0 | 213.34±95.4 | 0.0±0.0 | 226.52±74.11 | 1.7±2.21 |
| Ant-Run ε = 0.025 | PPOL-vanilla | 700.14±4.1 | 2.24±1.39 | 692.1±4.27 | 7.13±3.39 | 700.82±5.09 | 33.24±4.72 |
| | PPOL-random | 695.72±15.1 | 1.98±1.6 | 689.63±13.95 | 5.27±3.5 | 684.69±28.52 | 24.35±8.6 |
| | SA-PPOL | 698.9±12.38 | 1.11±1.22 | 697.87±12.21 | 3.03±2.37 | 700.42±10.18 | 33.17±10.09 |
| | SA-PPOL(MC) | 381.41±254.34 | 5.08±5.45 | 377.31±252.4 | 4.63±4.97 | 388.9±261.43 | 15.61±17.45 |
| | *SA-PPOL(MR) | 115.82±34.68 | 6.68±3.68 | 115.99±33.91 | 6.54±3.58 | 114.99±30.18 | 7.61±2.74 |
| | ADV-PPOL(MC) | 613.48±3.46 | 0.0±0.0 | 609.07±3.77 | 0.0±0.0 | 633.9±6.51 | 1.25±0.67 |
| | ADV-PPOL(MR) | 594.04±11.5 | 0.0±0.0 | 590.49±12.15 | 0.0±0.0 | 618.69±12.63 | 0.42±0.32 |
| Car-Circle ε = 0.05 | PPOL-vanilla | 404.36±8.03 | 29.56±12.99 | 181.03±31.97 | 4.91±24.06 | 330.53±8.4 | 30.91±7.38 |
| | PPOL-random | 412.98±9.32 | 2.14±3.96 | 337.48±12.51 | 103.52±12.16 | 370.9±8.83 | 52.28±5.95 |
| | SA-PPOL | 423.64±11.11 | 5.12±9.18 | 324.03±52.9 | 17.38±40.42 | 396.37±18.7 | 41.8±10.83 |
| | SA-PPOL(MC) | 372.04±18.55 | 0.09±0.59 | 192.73±13.41 | 0.04±0.42 | 353.1±13.2 | 29.86±4.01 |
| | SA-PPOL(MR) | 376.48±31.37 | 0.2±0.99 | 261.53±62.04 | 0.11±0.79 | 367.92±38.05 | 30.69±5.34 |
| | ADV-PPOL(MC) | 270.0±17.19 | 0.01±0.19 | 319.08±20.44 | 4.4±8.53 | 274.12±6.87 | 2.47±2.06 |
| | ADV-PPOL(MR) | 273.16±19.49 | 0.0±0.0 | 318.12±20.57 | 0.08±0.87 | 274.68±9.63 | 0.66±1.08 |
| Drone-Circle ε = 0.025 | PPOL-vanilla | 658.74±103.47 | 14.26±9.55 | 265.11±188.91 | 42.04±37.29 | 423.39±60.07 | 22.34±10.33 |
| | PPOL-random | 713.52±51.27 | 7.26±8.48 | 213.57±163.64 | 32.79±32.07 | 446.03±54.79 | 17.62±8.77 |
| | SA-PPOL | 594.56±55.41 | 3.16±4.0 | 522.52±133.0 | 23.64±18.18 | 456.11±56.71 | 20.62±10.83 |
| | SA-PPOL(MC) | 465.65±94.17 | 2.06±5.1 | 350.19±113.96 | 1.91±5.18 | 396.93±66.09 | 16.55±7.52 |
| | SA-PPOL(MR) | 330.77±145.12 | 3.17±6.49 | 292.4±132.96 | 2.6±6.57 | 301.27±131.61 | 19.23±10.92 |
| | ADV-PPOL(MC) | 302.79±65.62 | 0.0±0.0 | 313.51±94.38 | 7.64±12.8 | 308.38±44.16 | 3.99±6.38 |
| | ADV-PPOL(MR) | 358.49±37.84 | 0.03±0.32 | 315.41±85.2 | 12.33±15.66 | 340.9±37.79 | 3.74±3.81 |
| Ant-Circle ε = 0.025 | PPOL-vanilla | 184.43±23.32 | 4.86±8.59 | 178.28±26.02 | 4.9±8.97 | 192.15±21.57 | 27.92±6.76 |
| | PPOL-random | 144.05±20.09 | 3.68±10.13 | 140.62±17.88 | 4.3±10.07 | 143.87±14.41 | 17.67±8.87 |
| | SA-PPOL | 193.81±35.02 | 2.78±7.25 | 191.61±30.14 | 3.5±7.56 | 204.17±27.43 | 28.74±7.52 |
| | *SA-PPOL(MC) | 0.6±0.43 | 0.0±0.0 | 0.62±0.41 | 0.0±0.0 | 0.63±0.33 | 0.0±0.0 |
| | *SA-PPOL(MR) | 0.63±0.42 | 0.0±0.0 | 0.61±0.43 | 0.0±0.0 | 0.62±0.35 | 0.0±0.0 |
| | ADV-PPOL(MC) | 120.54±20.63 | 1.36±5.27 | 118.68±23.49 | 1.58±6.66 | 119.2±13.99 | 2.68±3.98 |
| | ADV-PPOL(MR) | 122.52±20.74 | 0.49±2.38 | 120.32±20.08 | 0.54±2.57 | 121.08±15.06 | 1.82±2.51 |

The experiment results of the maximum-entropy method: SAC-Lagrangian is shown in Table 6. We evaluated the effect of different entropy regularizers $\alpha$ on the robustness against observational perturbation. Although the trained agents can achieve almost zero constraint violations in noise-free environments, they suffer from vulnerability issues under the proposed MC and MR attacks. Increasing the entropy cannot make the agent more robust against adversarial attacks.

Table 6: Evaluation results of natural performance (no attack) and under MC and MR attackers of SAC-Lagrangian w.r.t different entropy regularizer $\alpha$. Each value is reported as: mean ± standard deviation for 50 episodes and 5 seeds.

| Env | $\alpha$ | Natural | | MC | | MR | |
| --- | --- | --- | --- | --- | --- | --- | --- |
| | | Reward | Cost | Reward | Cost | Reward | Cost |
| Car-Circle $\epsilon = 0.05$ | 0.1 | 414.43±7.99 | 1.04±2.07 | 342.32±17.8 | 112.53±6.92 | 328.5±22.06 | 43.52±18.39 |
| | 0.01 | 437.12±9.83 | 0.94±1.96 | 309.0±60.72 | 92.53±22.04 | 313.58±21.1 | 35.0±15.59 |
| | 0.001 | 437.41±10.0 | 1.15±2.36 | 261.1±53.0 | 65.92±24.37 | 383.09±50.06 | 53.92±16.3 |
| | 0.0001 | 369.79±130.6 | 5.23±11.13 | 276.32±107.11 | 84.21±35.68 | 347.85±117.79 | 52.97±22.4 |
| Car-Run $\epsilon = 0.05$ | 0.1 | 544.77±17.44 | 0.32±0.71 | 599.54±10.08 | 167.77±29.44 | 591.7±27.82 | 158.71±51.95 |
| | 0.01 | 521.12±23.42 | 0.19±0.5 | 549.43±53.71 | 73.31±54.43 | 535.99±23.34 | 21.71±24.25 |
| | 0.001 | 516.22±47.14 | 0.47±0.95 | 550.29±34.78 | 90.47±48.81 | 546.3±54.49 | 87.25±60.46 |
| | 0.0001 | 434.92±136.81 | 0.0±0.0 | 446.44±151.74 | 41.89±44.41 | 452.29±119.77 | 15.16±17.7 |

**Linearly combined MC and MR attacker.** The experiment results of trained safe RL policies under the mixture of MC and MR attackers are shown in Figure 5 and some detailed results are shown in Table 7. The mixed attacker is computed as the linear combination of MC and MR objectives, namely, $w \times MC + (1 - w) \times MR$, where $w \in [0, 1]$ is the weight. Our agent (ADV-PPOL) with adversarial training is robust against the mixture attacker. However, there is no obvious trend to show which weight performs the best attack. In addition, we believe the performance in practice is heavily dependent on the quality of the learned reward and cost Q functions. If the reward Q function is learned to be more robust and accurate than the cost Q function, then giving larger weight to the reward Q should achieve better results, and vice versa.

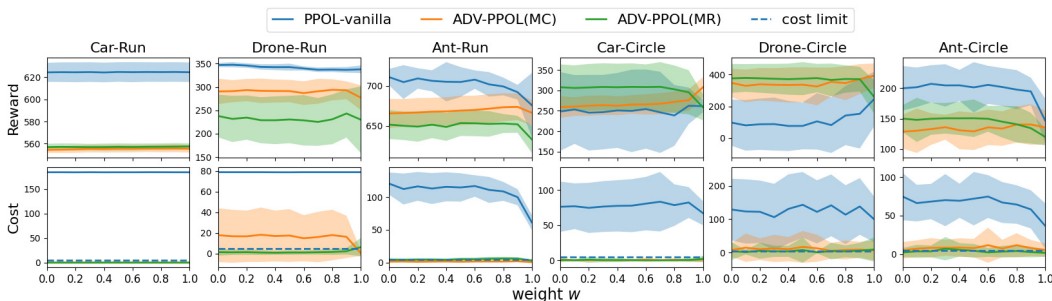

Figure 5: Reward and cost of mixture attackers of MC and MR

Table 7: Evaluation results under different ratio of MC and MR attackers of PPOL. Each value is reported as: mean ± standard deviation for 50 episodes and 5 seeds.

| Env | Method | MC:MR=3:1 | | MC:MR=1:1 | | MC:MR=1:3 | |
| --- | --- | --- | --- | --- | --- | --- | --- |
| | | Reward | Cost | Reward | Cost | Reward | Cost |
| Car-Circle $\epsilon = 0.05$ | PPOL-vanilla | 247.75±92.34 | 84.68±40.6 | 253.11±96.46 | 78.25±35.53 | 241.91±98.26 | 75.22±38.55 |
| | ADV-PPOL(MC) | 268.6±26.79 | 0.7±3.13 | 261.98±26.12 | 0.18±1.27 | 262.14±23.26 | 0.77±3.83 |
| | ADV-PPOL(MR) | 307.76±58.0 | 0.08±0.8 | 306.8±56.03 | 0.39±2.54 | 308.17±57.43 | 0.6±3.5 |
| Car-Run $\epsilon = 0.05$ | PPOL-vanilla | 624.3±8.55 | 183.95±0.43 | 624.68±8.85 | 184.22±0.45 | 624.54±8.89 | 184.03±0.41 |
| | ADV-PPOL(MC) | 555.45±3.37 | 0.03±0.18 | 555.64±3.47 | 0.02±0.13 | 555.22±3.09 | 0.0±0.0 |
| | ADV-PPOL(MR) | 557.05±3.06 | 0.02±0.13 | 557.07±3.05 | 0.08±0.28 | 556.9±2.96 | 0.08±0.28 |

Table 8: Evaluation results of natural performance (no attack) and under MAD, MC, and MR attackers of CVPO. Each value is reported as: mean ± standard deviation for 50 episodes and 5 seeds.

| Env | Natural | | MAD | | MC | | MR | |
|---|---|---|---|---|---|---|---|---|
| | Reward | Cost | Reward | Cost | Reward | Cost | Reward | Cost |
| Car-Circle $\epsilon = 0.05$ | 412.17±13.02 | 0.02±0.13 | 236.93±72.79 | 49.32±34.01 | 310.64±37.37 | 98.03±25.53 | 329.68±77.66 | 51.52±24.65 |
| Car-Run $\epsilon = 0.05$ | 530.04±1.61 | 0.02±0.16 | 481.53±17.43 | 2.55±3.11 | 537.51±8.7 | 23.18±16.26 | 533.52±2.87 | 14.42±6.77 |

The experiments results of CVPO (Liu et al., 2022) is shown in Table 8. We can that the vanilla version is not robust against adversarial attackers since the cost is much larger after being attacked. Based on the conducted experiments of SAC-Lagrangian, FOCOPS, and CVPO, we can conclude that the vanilla version of them all suffer from vulnerability issues: though they are safe in noise-free environments, they are no longer safe under strong MC and MR attacks, which validate that our proposed methods and theories could be applied to a general safe RL setting.

