# OpenReview forum: "On the Robustness of Safe Reinforcement Learning under Observational Perturbations"
_ICLR.cc/2023/Conference — ICLR 2023 poster_

### Official Review · Reviewer_NTgs · 2022-10-24

**Confidence:** 3
**Correctness:** 3
**Technical Novelty And Significance:** 2
**Empirical Novelty And Significance:** 2
**Recommendation:** 6

**Clarity, Quality, Novelty And Reproducibility:**

Clarity: The problem of adversarial attack in the CMDP setting is not clearly stated. In addition, it would be nice to better interpret the theoretical guantee. Why is an upper bound instead of a lower bound of constraint violation interesting?

Novelty: It would be great to have an in-depth comparison to existing works on adversarial attacks in MDP in terms of theory and algorithm.

**Strength And Weaknesses:**

Strength:
This paper conducted thorough numerical experiments.

Weaknesses:

1. I feel that the problem of adversarial attack of the CMDP problem is not well motivated. The reason is that MDP is a special case of the CMDP problem. As a result, any adversarial attack method for MDP can be used to attack CMDP. Thus, it would be interesting to see if there is any separation between these two problem classes. Moreover, the problem setting should be presented more clearly, with some rigorous statement of the problem. What is the objective of adversarial attacks in CMDP? Do we want to let the policy be suboptimal or do we want to let it violate the safety constraint?

2. It seems that the problem of adversarial attack has been extensively studied in the literature and this work seems a bit derivative given the advancement in this strand of research. For example, Theorem 2 in this work seems quite similar to Theorem 6 in Zhang et al 2020.

3. The adversarial training algorithm is only heuristic and there is no theoretical guarantees.

**Summary Of The Paper:**

This paper studies the problem of adversarial attacks of constrained MDP. The authors propose two adversarial attack approaches with one maximizing the cost and the other maximizing the reward. The authors also establish an upper bound on constraint violations of the adversary attack. Furthermore, a heuristic algorithm of adversarial training that aims to achieve robustness is proposed.

**Summary Of The Review:**

The problem setting and research goal should be explained in detail and well motivated. Novelty of this work needs to be better highlighted.

---

> ### Author Response · Authors · 2022-11-08
> **Response to Reviewer NTgs (Part 2/2)**
>
> (continued)
>
> > Q2: It seems that the problem of adversarial attack has been extensively studied and this work seems a bit derivative given the advancement in this strand of research. For example, Theorem 2 in this work seems quite similar to Theorem 6 in Zhang et al 2020.
>
> Theorem 2 in our paper gives the cost value bound under a one-step attack:
>
> >> $$ \tilde{V}^{\pi,\nu}_ c (s) - V_ c^{\pi}(s) \leq 2L\epsilon \left(p_ s C_ m + \frac{\gamma C_ m}{1-\gamma}\right). $$
>
> Theorem 6 in Zhang et al 2020 states how to compute the TV distance between two policies:
>
> >> $$D_ {TV}(\bar{\pi}(\cdot|s), \bar{\pi}(\cdot | \hat{s})) = \sqrt{2/\pi} \frac{d}{\sigma} + O(d^3), d=|| \pi(s) - \pi(\hat{s})||_ 2.$$
>
> Apologize if we miss anything but we are really confused about the reviewer's criticism as the two theorems have different formats, goals, and conclusions. We really cannot figure out the similarity between them.
>
> Our conjecture is that the reviewer would concern that adversarial attacks have been well-studied. We want to emphasize that our work on safe RL significantly differs from previous works. We detail them as follows:
>
> 1. It is novel to propose the attacks and the robust training scheme, as we can hardly find effective ones for safe RL in the literature.
> As we answered in Q1, the distinction between MDP and CMDP makes the attacking objective fundamentally different, and thus the adversarial attack methods in MDP are not suitable in CMDP.
> Also, the robust training techniques that worked well previously can not improve safety in safe RL.
> Therefore, we believe our findings of the vulnerability of safe RL under the maximum-reward/cost attacks and the adversarial training method are novel, as we suggest the existence of a previously unrecognized problem in safe RL.
>
> 2. The definitions and theoretical results in Sec. 3 are new and important.
> In previous works, we rarely see formal discussions of them.
> For instance, the temptation definition and Lemma 1 can explain why safe RL methods cannot achieve both the highest reward and the lowest cost due to the inherent trade-offs in CMDP.
> Lemma 2 explains why the cost curves in many safe RL papers finally converge to the threshold.
> Those observations widely exist in the safe RL literature but with limited discussions, and sometimes may even confuse the readers in that "why a method cannot have the highest reward return as well as satisfying the constraints?"
> With the theories we provided, those confusions are resolved.
>
>
> 3. The theoretical analysis of Theorem 2 regarding the cost value bound is original and novel.
> It might look similar to Theorem 5 in Zhang et al. 2020, but in fact, they are quite different. The previous bound is entirely determined by the TV distance between the natural policy and the corrupted policy, while ours decouples the dependency to 1) the smoothness of the policy $L$, 2) the perturbation range $\epsilon$, and 3) the probability of entering unsafe states $p_s$. We show that the bound is tight.
> Theorems 2 and 3 suggest that a smoother policy has better robustness, which bridges some conclusions in adversarial machine learning with robust safe RL, and we haven't seen similar claims in the RL literature.
>
> Therefore, we do believe that our original contributions and novelty could improve the awareness of the robust safe RL topic and benefit the community.
>
> > Q3: The adversarial training algorithm is only heuristic and there are no theoretical guarantees.
>
> We humbly can not agree that our adversarial training method is purely heuristic-based.
> In fact, **it is theoretically backed up by Theorem 4 and Proposition 2 in Sec. 4.1**.
> The theoretical analysis is non-trivial. Due to the page limit, we put the most important theoretical results in the main paper and leave the detailed analysis and proofs in Appendix A.6.
> We highlight the theory support as follows.
>
> - In Theorem 4, we first show why we can perform adversarial training. Since the Bellman operator is a contraction for both reward and cost _under observational adversaries_, we can accurately evaluate the performance of the _corrupted policy_ in the policy evaluation phase.
> This is a crucial and strong guarantee for the success of adversarial training, because we can not improve the policy without well-estimated values.
>
> - In Proposition 2, we provide theoretical guarantees of the safety performance of adversarial training under the MC and MR attackers.
> The adversarial trained agents under the proposed attacks are guaranteed to be safe under any bounded adversarial perturbations.
> We believe the above theoretical guarantees are crucial for the success of our adversarial training agents, because from our ablation studies of using other attackers (MAD or random noise), we can see adversarial training can not achieve desired performance with other attackers.
>
> We hope our response can fully address the reviewer’s concern, and we are glad to discuss if there are further questions.

---

> > ### Comment · Reviewer_NTgs · 2022-12-05
> > **followup comment**
> >
> > Here I meant Theorem 5 in Zhang et al. Sorry for the typo. It seems that the analysis of Theorem 2 is based on modifying standard analysis of approximate dynamic programming and is similar to that of Theorem 5 in Zhang et al.
> >
> > Theorem 4 and Proposition 2 seem insufficient to be called the theory of the proposed algorithm. Given these results, there are still a few outstanding questions: what is the sample and iteration complexity of the algorithm? If the algorithm is run only by a finite number of iterations, can robustness approximately hold?

---

> > > ### Author Response · Authors · 2022-12-05
> > > **Response to reviewer NTgs regarding the follow-up comments**
> > >
> > > We thank the reviewer for your reply and additional comments. We address the questions as follows.
> > >
> > > > Q1: It seems that the analysis of Theorem 2 is based on modifying standard analysis of approximate dynamic programming and is similar to that of Theorem 5 in Zhang et al.
> > >
> > > Theorem 2 in our paper gives the cost value bound for **any** state $s$ under **any** adversary:
> > >
> > > >> $$\tilde{V}^{\pi,\nu}_ c (s) - V_ c^{\pi}(s) \leq 2L\epsilon \left(p_ s C_ m + \frac{\gamma C_ m}{1-\gamma}\right). $$
> > >
> > > Theorem 5 in Zhang et al. gives the cost bound for the worst-case state $s \in \mathcal{S}$ under the **minimizing-reward** adversary:
> > >
> > > >> $$ \max_ {s\in\mathcal{S}} V^\pi(s) -\tilde{V}^{\pi \circ \nu} (s) \leq \alpha \max_ {s\in\mathcal{S}} \max_ {\tilde{s}\in B(s)} D_ {TV}(\pi(\cdot|s), \pi(\cdot|\tilde{s})), \quad \alpha= 2[1+ \frac{\gamma}{(1-\gamma)^2}]R_ {max}$$
> > >
> > > In fact, our Theorem 2 is quite different from the Theorem 5 in Zhang et al. from several aspects.
> > >
> > > First, our bound holds for **any** state under **any** adversaries, while the previous bound in Zhang et al. only holds for the **worst-case** state under the **minimizing reward** attack. Therefore, our result is more general to the observational attack setting in safe RL.
> > >
> > > Second, the previous bound is entirely determined by the TV distance between the natural policy and the corrupted policy, while ours decouples the dependency to 1) the smoothness of the policy $L$, 2) the perturbation range $\epsilon$, and 3) the probability of entering unsafe states $p_s$. We show that the bound is tight.
> > > Our results suggest that a smoother policy has better robustness, which bridges some conclusions in adversarial machine learning with robust safe RL, and we haven't seen similar claims in the RL literature.
> > >
> > > Therefore, we do believe that our original contributions and novelty could improve the awareness of the robust safe RL topic and benefit the community.
> > >
> > > > Q2: What is the sample and iteration complexity of the algorithm?
> > >
> > >
> > > Note that Algorithm 1 is essentially updating the corrupted policy $\pi \circ \nu$ and its critics $V^{\pi \circ \nu}$, which has similar complexity to other actor-critic-based RL algorithms.
> > > The exact complexity analysis depends on the choice of the learner and representations ([1](https://arxiv.org/abs/2004.12956), [2](https://arxiv.org/abs/1910.08412), [3](https://homes.cs.washington.edu/~sham/papers/thesis/sham_thesis.pdf)), which is far beyond the scope of our paper.
> > >
> > >
> > > > Q3: If the algorithm is run only by a finite number of iterations, can robustness approximately hold?
> > >
> > > As mentioned in Q2, the proposed algorithm shares similar convergence analysis to other actor-critic-based RL algorithms.
> > > Since we use neural network representations of the policy and critics, it is usually hard to give a generic exact bound for $V^{\pi\circ\nu}$ after a certain number of iterations, but we can draw some conclusions like
> > >
> > > >> If the performance after $k$-th iteration is $V^{\pi_ k\circ\nu_ k}$, the performance after $(k+1)$-th iteration satisfies $V^{\pi_ {k+1}\circ\nu_ {k+1}}\leq V^{\pi_ k\circ\nu_ k} + C$, where $C$ is a term related to the distance between $\pi_ k\circ\nu_ k$ and $\pi_ {k+1}\circ\nu_ {k+1}$
> > > (by Corollary 1 in [4](https://arxiv.org/pdf/1705.10528.pdf)).
> > >
> > > Note that such analysis may differ according to the safe RL `Learner`.
> > > In practice, our experiments show that the algorithm with the `PPO-Lagrangian` learner can converge and achieve robustness on a variety of safe RL tasks within finite iterations.
> > > We believe that more rigorous convergence analysis for a finite number of iterations requires stronger assumptions of the problem setting, which could be a future direction based on our work.
> > >
> > >
> > > We hope our response can address the reviewer’s concern, and we are more than willing to discuss if there are further questions.

---

> > > > ### Comment · Reviewer_NTgs · 2022-12-13
> > > > **Re**
> > > >
> > > > I would like to thank the authors for their response. I have raised the score to 6.

---

> ### Author Response · Authors · 2022-11-08
> **Response to Reviewer NTgs (Part 1/2)**
>
> We thank the reviewer for your time and valuable feedback. We address the concerns as follows.
>
> > Q1.1: I feel that the problem of adversarial attack of the CMDP problem is not well motivated. The reason is that MDP is a special case of the CMDP problem. As a result, any adversarial attack method for MDP can be used to attack CMDP. Thus, it would be interesting to see if there is any separation between these two problem classes.
>
> The question is exactly what motivates our work.
> In fact, most people have the same intuition as the reviewer: it seems that we can directly apply the methods in MDP to CMDP.
> However, as we presented in Sec. 5.1 and 5.2, it is very risky to believe so because **the SOTA attacks (MAD, minimize reward) and defense (SA-PPOL, adding random noise) methods in MDP do not work well in CMDP**. We hope we had made it clear in the abstract: `we formally analyze the unique properties of designing adversarial attackers in the safe RL setting. We show that attack techniques for standard RL are ineffective for safe RL and thus propose two new approaches.`
>
> We can illustrate this with the Ant-Run experiment. We summarize the **constraint violations** of MDP methods and our methods for CMDP in the following table, where $\uparrow$ and $\downarrow$ means the increase and decrease when compared to no attacking case.
> From the first two columns,
> we can see SOTA attacks in MDP fail to induce risky behaviors in CMDP, while our attacking methods effectively increase constraint violations which is important in helping designing defense algorithms.
> Specifically, the SA-PPOL method, which is the SOTA robust training method for MDP, will cause $\textbf{60}$ times more constraint violations than our proposed method under MC attacks.
> Therefore, **our paper is well-motivated**, since we found that the adversarial attack and training techniques for MDP can hardly be applied to CMDP.
> The failures of MDP methods are because of the unique properties in CMDP, as we detailed in Def. 1, Lemma 1, and Lemma 2: the optimal policy in CMDP is located on the safety boundary and can be easily tempted.
>
>
> | Attacker |       PPOL-random (MDP)     |         SA-PPOL   (MDP)     |     ADV-PPOL (ours)     |
> |:--------:|:----------------------:|:----------------------:|:--------------------:|
> |    MAD (MDP)  |  2.09($\uparrow$1.08)  |  0.75($\uparrow$0.29)  |  0.0($\uparrow$0.0)  |
> |   MinR (MDP)  |  0.0($\downarrow$1.01) |  0.0($\downarrow$0.46) | 0.0($\downarrow$0.0) |
> |    MC  (ours)  | 45.94($\uparrow$44.93) | 67.68($\uparrow$67.22) |  1.1($\uparrow$1.1)  |
> |    MR  (ours)  | 46.97($\uparrow$45.96) |  87.1($\uparrow$86.64) | 1.75($\uparrow$1.75) |
>
> We also highlight the reason for failure in Sec. 5.1: `MAD fails to perform an effective attack in increasing the cost because the reward decrease keeps the agent away from high-risk regions`, which is also shown in our attached video on the project website. The results show how the SOTA attacks in MDP fail in CMDP and how unsafe it is to use the robust training method for MDP.
>
> We revised and highlighted the paper's novelty, problem statement, and contributions in blue for your reference.
>
> > Q1.2 The problem setting should be presented more clearly, with some rigorous statement of the problem. What is the objective of adversarial attacks in CMDP? Do we want to let the policy be suboptimal or do we want to let it violate the safety constraint?
>
> The objective of attacking in safe RL is to increase constraint violations and keep stealthy.
> As we stated in Sec. 1: `the constraint violation cost should be more important than the measure of reward`, since `any constraint violations could be fatal and unacceptable in the real world`.
> Therefore, the objective of attacks in CMDP is to induce unsafe behaviors while staying unrecognized as much as possible.
> We revised Sec. 3.2 and Sec. 4.1 with rigorous statements of attacking and robust training in safe RL (highlighted in Blue).
> We also discussed how to evaluate attacks for safe RL in Appendix B.1.
>
> (continued)

---

### Official Review · Reviewer_GBM8 · 2022-10-24

**Confidence:** 4
**Correctness:** 3
**Technical Novelty And Significance:** 3
**Empirical Novelty And Significance:** 3
**Recommendation:** 6

**Clarity, Quality, Novelty And Reproducibility:**

The paper is clearly written by addressing an important problem. The authors proposed some novel idea in the robustness of safe RL.

**Strength And Weaknesses:**

Strong points:
This work first considers state observation disturbance of safe RL, and proposes the attack strategy to make the attacked policy with high return and high cost, which are called as Stealthiness and Effectiveness of attacking methods in this paper.

This work proposes a maximum reward attacker to make the victim policy infeasible by increasing the return of the policy with theoretical guarantee. Also, this work proposes corresponding adversarial training algorithm. Experimental results show the effectiveness of the attack and defence algorithms.

Weak points:
In the proof of Lemma 2 (Appendix A.2), this paper conducts the policy $\bar{\pi}$ by $\pi’$ and $\pi^*$, and shows that $\bar{\pi}$. However, the description of $\bar{\pi}\in\Pi$ is missing and it seems that $\Pi$ needs a kind of property of convexity, which should be clarified in the result and the proof.
Also in the proof of Lemma 2, I’m a little confused about Eq.(10), this equation derives the distribution of trajectories $\tau$ rather than the expression of the policy $\bar{\pi}$, can the authors explain more about the form of the policy $\bar{\pi}$ via $\pi’$ and $\pi^*$?

In the proof of Proposition 1, this paper claims that the policy $\pi^* \circ v_{MR}$ is within the tempting policy class without showing that it is in the policy space $\Pi$. And more discussion about properties of $\Pi$ will be beneficial.

**Summary Of The Paper:**

This paper first formally analyze the vulnerability of optimal policies trained by safe RL algorithms under observational disturbance. By defining state-adversarial safe RL, this paper theoretically shows these optimal policies are vulnerable under observational adversarial attacks. Based on these analyses, this paper proposes two kinds of attacking algorithm, maximizing the constraint violation cost, and maximizing the task reward to induce a tempting but risky policy, as well as an adversarial training algorithm. Experimental results show the effectiveness of their methods.

**Summary Of The Review:**

Overall, I am leaning toward recommending weak accept and I think some further explanation of $\Pi$ can make this article more solid.

---

> ### Author Response · Authors · 2022-11-08
> **Response to Reviewer GBM8**
>
> We thank the reviewer for acknowledging our work and valuable feedback. We clarify the policy space definition as follows.
>
> First, we agree that the policy space $\Pi$ definition is missing in the proof and we have added and highlighted it in our revision.
> We consider the policy space $\Pi:\mathcal{S}\times\mathcal{A} \to [0,1]$.
> The tempting policy space and the feasible policy space are subsets of the policy space: $\Pi^T \subset \Pi$, $\Pi^\kappa \subset \Pi$ by definitions 1 and 3.
> Then we can show that any corrupted policy $\pi \circ \nu$ is within the policy space $\Pi$, because $\nu(s) \in \mathcal{S}$ and thus $\pi \circ \nu = \\{\pi(a|\nu(s)): \forall a \in \mathcal{A}, \forall s \in \mathcal{S} \\} \in \Pi$.
> Therefore, the optimal policy under the MR attack $\pi^* \circ \nu_{MR}$ is also in the policy space $\Pi$, and since it satisfies the condition of the tempting policy, it belongs to the temping policy class $\Pi^T$.
>
>
> Second, we provide a more rigorous claim for Lemma 2 by leveraging the `option` learning framework.
> We enlarge the policy space $\Pi$ to be an augmented space $\bar{\Pi}: \mathcal{S}\times\mathcal{A}\times\mathcal{O}\rightarrow [0,1)$, where $\mathcal{S}, \mathcal{A}$ denote the state and action space and $\mathcal{O}=\\{0,1\\}$ is the indicator space.
> This definition of policy space is commonly used in hierarchical RL and can be viewed as a subset of option-based RL [[Sutton, 2019](https://www.sciencedirect.com/science/article/pii/S0004370299000521), [Riemer, 2018](https://arxiv.org/abs/1810.11583), [Zhang, 2019](https://arxiv.org/pdf/1904.12691.pdf)] ($\mathcal{O}$ corresponds to the `option` space).
>
>
> With this definition, we can represent policy $\bar{\pi} \in \bar{\Pi}$ as $\bar{\pi}(a|s,o)$ as
>
> $$
>   \bar{\pi}(a_ t|s_ t,o_ t)=
>   \pi'(a_ t|s_ t), \text{ if } o_ t=1, \quad \bar{\pi}(a_ t|s_ t,o_ t)= \pi^*(a_ t|s_ t), \text{ if } o_ t=0
> $$
> with $o_ {t+1}=o_ t,o_ 0 \sim \text{Bernoulli}(\alpha)$. In this case, eq. (12)(13) still hold and the proof is valid.
>
>
> We use the enlarged policy space to demonstrate and prove Lemma 2, however, we want to point out that it is usually **not a necessary condition** in many safe RL problems with continuous state and action spaces because the metric space would be a connected set, which is also the reason why we see the safe RL policies in $\Pi$ usually converge to the constraint threshold $\kappa$ in many previous papers.
> We have updated Sec. 3.1, Sec. 3.3, and Appendix A.2 with more rigorous statements and above discussions, which are highlighted in **purple color**.
>
> We sincerely appreciate the reviewer for the constructive comments to make our paper more solid, and we are more than glad to address if there are any other questions.

---

### Official Review · Reviewer_7p6W · 2022-10-25

**Confidence:** 3
**Correctness:** 4
**Technical Novelty And Significance:** 2
**Empirical Novelty And Significance:** 2
**Recommendation:** 6

**Clarity, Quality, Novelty And Reproducibility:**

The investigation of observational adversarial attacks for safe RL is novel. The proposed training algorithm lacks a detailed description. The authors provide the code, which enhances the reproducibility of their experiments.

**Strength And Weaknesses:**

Strengths:

1. The investigation of observational adversarial attacks for safe RL is novel.
2. This paper provides comprehensive experimental results to support their theoretical findings.

Weaknesses:

1. While the formulation of observational adversarial attacks for safe RL is novel compared to the safe RL literature, it is unclear to me if such state adversarial attacks well model the observational perturbations in real-world applications. For example, in the Bellman equation for safe RL with observational adversarial attacks (Eq. 6), the observational adversary $\nu(s)$ is the same for each step. I think in real-world applications, the adversarial attacks can be different at different steps.
2. Since the proposed two adversarial attacks, i.e., maximum reward attacker and maximum cost attacker are defined based on taking the maximum over all possible observational adversaries $\nu(s): \mathcal{S} \mapsto \mathcal{S}$. How large is the space of the observational adversary $\nu$? It is unclear to me how to efficiently find the maximum reward attacker and maximum cost attacker over all possible attackers, both theoretically and empirically?
3. The proposed training algorithm for safe RL (Algorithm 1) needs a more detailed description and a connection with the theoretical findings in Section 3. For example, the authors should give a more detailed discussion on how “scheduler” updates the adversary $\nu$ and finds the maximum reward attacker and maximum cost attacker.


**Summary Of The Paper:**

This paper shows that the optimal solutions of many safe RL problems are not robust and safe against observational perturbations. Instead, this paper investigates the unique properties of effective state adversarial attackers for safe RL. The authors design two observational perturbations, i.e., one maximizes the cost and the other maximizes the reward. The authors design an adversarial training framework for safe RL and validate its effectiveness by experimental results.

**Summary Of The Review:**

This paper studies an interesting and novel problem, i.e., safe RL under observational adversarial attacks. But this paper only considers a fixed adversary for each step, which may not fit the real-world applications with adversarial attacks. Also, it is unclear to me how to efficiently find the designed maximum reward attacker and maximum cost attacker from all possible observational adversaries. The proposed training algorithm also lacks a detailed description. Due to the above reasons, I give borderline rejection.

If the authors can well address my concerns, I am ready to raise my score.

====

Thank the authors for their response.

For the motivation, I think that compared to using a function $\nu(\cdot)$ to formulate the adversary, using a fully adversarial way to formulate the adversary (e.g., considering an adversarial corrupted state $s_t$) may better fit the possible adversarial attacks in real-world RL applications. Since the function $\nu(\cdot)$ is also fixed throughout the whole RL game, I am not sure if it is very suitable for complicated real-world adversarial attack applications (although I understand that $\mu(\cdot)$ makes the problem more tractable).

My concerns on the implementation of the proposed methods are addressed. I raise my score from 5 to 6, but I do not have a very high confidence and would like to listen to the opinions from other reviewers and AC.

---

> ### Author Response · Authors · 2022-11-08
> **Response to Reviewer 7p6w**
>
> We thank the reviewer for your time and valuable feedback. We address the concerns as follows.
>
> > Q1: I think in real-world applications, the adversarial attacks can be different at different steps.
>
> We understand the reviewer's concern and apologize if our notation of the adversary confused the reviewer.
> We clarify that **this is a misunderstanding** and **the adversarial state can be different at different steps** in our formulation.
> The adversary $\nu$ is a function that maps a state to another state, while $\nu(s)$ denotes the corrupted state of state $s$.
> In our setting, though the function $\nu$ is supposed to be fixed for each training iteration to facilitate analysis, **the corrupted state $\nu(s)$ is not the same and is different based on different $s$**.
> Writing it as $\nu({s_ t})$ may help avoid potential confusion.
> At each timestep, the corrupted state $\nu(s_ t)$ is different at different steps since $s_ t$ will change, which models real-world applications well.
> In addition, the proposed MR and MC attackers are solved based on the Q-value functions of the policy, which means that when the policy is getting better, the attackers will also be stronger. We detail this in the following answers.
>
> > Q2: How to efficiently solve the MC and MR attackers, both theoretically and empirically?
>
> We thank the reviewer for asking the question.
> Let's take the MC attacker as an example, the same analysis applies to MR as well.
> Theoretically, solving the MC attacker is equivalent to solving another MDP $\tilde{M}$, see the proof of Theorem 1.
> Intuitively, we can view the MC attacker as an agent whose observation is the same as the safe RL agent's, but the output action is the corrupted state.
> The reward and transition in $\tilde{M}$ depend on the safe RL agent's policy $\pi$.
> So in theory, the best way to find the MC attacker is to solve $\tilde{M}$ using standard RL algorithms for every updated safe RL policy.
> However, this requires a lot of computational burdens and is time-consuming.
>
> In practice, we note that we can approximate the MC objective by using the Q value estimation $Q_ c^{\pi}$ of the policy $\pi$, as shown in Equation (2) in the paper.
> Then for the MC attacker, we obtain the corrupted state $\tilde{s}$ for state $s$ by maximizing the loss:
> $\max _ {\tilde{s} \in B^\epsilon_ p(s)} \mathbb{E}_ {\tilde{a} \sim \pi(a|\tilde{s})} \left[ Q_ c^{\pi}(s, \tilde{a})) \right]$.
>
> Since the Q-value function and policy are parametrized by neural networks,
> we can backpropagate the gradient from $Q_ c^{\pi}(s, \tilde{a})$ to $\tilde{s}$ via $\pi(\tilde{a} |\tilde{s})$, which can be solved efficiently by many optimizers like ADAM.
> It is related to the [Projected Gradient Descent (PGD) attack](https://arxiv.org/abs/1706.06083) in the literature.
> It is also similar to the deterministic policy gradient method such as DDPG and TD3, but the optimization variables are the state perturbations rather than the policy parameters.
> This technique for solving adversarial attacks is also widely used in the standard RL literature and is shown to be successful, such as [Zhang, 2020](https://arxiv.org/abs/2003.08938) and [Pattanaik, 2018](https://arxiv.org/abs/1712.03632).
> However, they focus on minimizing the reward objective rather than the MC and MR attackers in our paper.
> Empirically we also show that this gradient-based method converges fast with a few iterations and within 10ms, which greatly improves adversarial training efficiency.
> We add the results, implementation details, and related discussions of the MC and MR attackers in Appendix C.1.
> We also highlight the description of practical implementation of attackers in Sec. 3.3 with teal color.
>
>
> > Q3: More details about the adversarial training and the scheduler.
>
> The "rollout trajectories" and "learner" steps are connected with Theorem 4 (Bellman contractions) and Proposition 2, since we can evaluate the value functions of the policy under attacks, and use them to improve the policy such that the updated policy is robust against strongest attacks.
>
> In our implementation, the Scheduler has two components: 1) updating the Q-value functions such that the MC/MR attackers are updated correspondingly and 2) linearly increasing the perturbation range from 0 to $\epsilon$ to make the training more stable.
> The reason for 1) is suggested in Q2, since attacking by MC or MR is essentially solving an optimization problem based on the Q-value function.
> The increasing perturbation trick is also used in previous work (Zhang 2020).
> Due to the page limit, we leave the detailed training algorithm in Appendix C.3.
>
>
> We hope our clarification can address the reviewer's concern, and we are more than glad to address if there are any questions.

---

> > ### Comment · Reviewer_7p6W · 2022-11-22
> > **Response to the Rebuttal**
> >
> > Thank you for your detailed reply.
> >
> > For the motivation, I think that compared to using a function $\nu(\cdot)$ to formulate the adversary, using a fully adversarial way to formulate the adversary (e.g., considering an adversarial corrupted state $s_t$) may better fit the possible adversarial attacks in real-world RL applications. Since the function $\nu(\cdot)$ is also fixed throughout the whole RL game, I am not sure if it is very suitable for complicated real-world adversarial attack applications (although I understand that $\mu(\cdot)$ makes the problem more tractable).
> >
> > My concerns on the implementation of the proposed methods are addressed.  I raise my score from 5 to 6, but I do not have a very high confidence and would like to listen to the opinions from other reviewers and AC.

---

### Official Review · Reviewer_yCc1 · 2022-11-01

**Confidence:** 4
**Correctness:** 3
**Technical Novelty And Significance:** 2
**Empirical Novelty And Significance:** 2
**Recommendation:** 6

**Clarity, Quality, Novelty And Reproducibility:**

The paper is well-written and easy to follow. I believe the novelty of the paper lies within the analysis of the value bound under observational perturbation for tempting safe RL. However, it is not clear how to detect a tempting safe RL. The code is provided for reproducibility which is a plus. I have gone through the proofs in the appendix and find no major problem other than the above comments.

**Strength And Weaknesses:**

Strengths:
- Safe RL is an active area of research and the paper studies the robustness of safe RL under observational perturbation attack which can benefit the community.
- Theoretical bounds are established for the value function under adversarial attack.
- Numerical experiments are extensive to show the effectiveness of the attacker and the robustness of the proposed adversarial training procedure.

Weaknesses:
- I think the authors already restrict the problem to the temping safe RL so the fact that the MR attacker works well makes sense. A more important question here is how to detect that we are in a temping safe RL setting so that this paper can be applied.
- I wonder if the authors already tried the baseline where the attacker tries to minimize the rewards as I believe this is a more natural baseline given that the authors propose the maximum reward attacker.
- There are few places in the proofs that need clarification, see comments below.
- The maximum reward attacker is only applicable when the constraint violation is merged with the original reward as in definition 1. As there are different way to characterize "safe" RL, the approach might not be applicable in other characterization.

Minor comments:
- I guess the authors use state and observation interchangeably. If not, then the use of each term should be consistent throughout the text.
- In section A4, the author seems to use $V^{\pi,\nu}_c (s)$ vs $\tilde{V}^{\pi,\nu}_c (s)$ and $V^{\pi}_c (s)$ vs $\tilde{V}^{\pi}_c (s)$ interchangeable. I wonder if there is a typo here.
- Before equation (35), why there is a coefficient of $1/2$ in the definition of the TV distance?
- Also, in (35), why there is a coefficient of 2 before the TV distance?


**Summary Of The Paper:**

The paper studies the robustness of safe RL agent under observation adversarial attack, They consider the setting of temping safe RL problem (there exists a better policy without the safe constraint). The authors propose two type of attackers that maximizes reward and provide theoretical bound  of the (cost/reward) value function under advesarial attack. After that, they propose an on-policy adversarial training strategy to defense against observational perturbation attack for safe RL. Experiments conducted on available benchmar to illustrate the performance of the two proposed attackers as well as the robustness of the adversarial training procedure.

**Summary Of The Review:**

The paper does have contribution in analysis the value bound for observational perturbation attack in safe RL. I think it is not a surprise that the MR attack works well in this problem setting as the safe constraint is characterized by a penalty threshold and the authors focus on the tempting safe RL problems. Numerical experiments are extensive to illustrate the performance of proposed approaches.

---

> ### Author Response · Authors · 2022-11-08
> **Response to Reviewer yCc1(Part 2/2)**
>
> (continued)
>
> > Q4: The maximum reward attacker is only applicable when the constraint violation is merged with the original reward as in definition 1. As there are different ways to characterize "safe" RL, the approach might not be applicable to other characterizations.
>
> First, maybe our notations confuse the reviewer, definition 1 about feasibility is actually entirely based on the constraint violation cost, i.e., any policies that satisfy the constraint is feasible.
> Therefore, the cost is not merged with the reward, but rather, they and their value functions are separated and independent. We added related discussions about this commonly used safe RL setting in Appendix B.1.
>
> Second, we do agree with the reviewer that in some "safe" RL settings such as merging the cost into the reward with some hand-tuned coefficients, the MR attacker may not be applicable.
> However, as we show in Appendix B.1, merging them together lacks interpretability.
> Therefore, as we answered in Q1, a rich literature on safe RL adopts the tempting setting, where the cost and reward are separated and the MR attacker should be effective.
> Empirically, we also conducted experiments for various safe RL algorithms in Appendix C.9, including FOCOPS, SAC-Lagrangian, and CVPO. All the results validate our theoretical analysis -- the vanilla safe RL policies are not robust against adversarial attacks.
> Therefore, this paper's claims and theoretical results should be generalizable to many safe RL settings and different approaches in the literature.
>
>
> We hope our answers can address the reviewer's concern, and we are more than glad to address if there are any questions.

---

> ### Author Response · Authors · 2022-11-08
> **Response to Reviewer yCc1 (Part 1/2)**
>
> We thank the reviewer for your time and valuable feedback. We address the questions as follows.
>
> > Q1: How to detect a tempting safe RL.
>
> This is an interesting point.
> We can easily determine whether a safe RL is tempting by the following rule: if it can be solved by **standard** RL algorithms, i.e., maximizing the reward without considering safety constraints, then it is **not** a tempting safe RL problem, and vice versa.
> In fact, the majority of safe RL papers and benchmarks consider the tempting setting, such as [CPO](https://arxiv.org/abs/1705.10528), [PCPO](https://arxiv.org/abs/2010.03152), [FOCOPS](https://arxiv.org/abs/2002.06506), [CVPO](https://arxiv.org/abs/2201.11927), and [SafetyGym](https://cdn.openai.com/safexp-short.pdf).
> Moreover, in these papers, we could observe that standard RL methods consistently outperform safe RL methods in terms of reward, which is exactly the property of temptation. However, no further discussions or analyses are given in the literature.
> Therefore, we are the first to formally investigate this widely existing but rarely discussed phenomenon by introducing the temptation definition, which is one contribution of our work.
> We added related discussions in the updated Appendix B.1.
>
>
> > Q2: Minimizing reward attacker baseline.
>
> We thank the reviewer for pointing this out.
> As suggested, we added the experiments of the minimizing reward attack.
> The results are shown in the table below, where $\uparrow$ and $\downarrow$ represent the increase and decrease w.r.t the natural performance respectively.
>
> | Method | | Car-Run  | Drone-Run | Ant-Run | Ant-Circle |
> |:------------:|:------:|:-----------------------:|:--------------------------:|:------------------------:|:--------------------------:|
> | PPOL-vanilla | Reward | 496.65 ($\downarrow$64.68) | 265.06($\downarrow$82.11)  | 498.42($\downarrow$179.98) | 67.9($\downarrow$89.54) |
> | |  Cost  | 0.0($\downarrow$0.15) |  0.0($\downarrow$0.0) | 0.03($\downarrow$1.2)  | 1.17($\downarrow$1.53) |
> | ADV-PPOL(MC) | Reward | 491.95($\downarrow$33.81) |  211.16($\downarrow$62.24)  |  548.0($\downarrow$53.25) |  86.26($\downarrow$49.72) |
> | |  Cost  | 0.0($\downarrow$0.0) |  0.4($\uparrow$0.4) | 0.0($\downarrow$0.0) | 0.0($\downarrow$0.3)  |
> | ADV-PPOL(MR) | Reward | 491.48($\downarrow$34.45) |  214.25($\downarrow$19.06)  | 524.24($\downarrow$95.93) | 87.24($\downarrow$46.03) |
> | |  Cost  | 0.0($\downarrow$0.0) | 1.1($\uparrow$1.1) | 0.0($\downarrow$0.17) | 1.4($\uparrow$0.53) |
>
> We can see that although it can reduce the agent’s reward quite well, it fails to increase the cost because the reward decrease may keep the agent away from high-risk regions.
> Note that the MAD attacker in the paper is designed for standard RL, which performs similarly to the minimizing reward attacker (Sec. 5.1).
> Therefore, an important and counter-intuitive finding is that the SOTA attacking methods for standard RL settings can hardly perform well in inducing unsafe behaviors for safe RL, which motivates us to propose the max-cost and max-reward attacking methods.
> Particularly, the max-reward attack is surprising because it is the objective in standard RL, which has never been viewed as an attacking objective in the literature.
> We add related discussions in Appendix C.9.
>
> > Q3.1: State and observation terms usage.
>
> Yes, we use them interchangeably since the CMDP is derived from the fully observable MDP. We agree that we should be consistent about the terms, so we have updated the paper, where we only use the observation term when an attack is applied.
>
> > Q3.2: $V_c^{\pi, \nu}, \tilde{V}_c^{\pi, \nu}, V_c^{\pi}, \tilde{V}_c^{\pi}$ usage.
>
> Yes, there are some typos in Eq.(33) and (35), and we have corrected them. Thanks for pointing them out.
> For further clarification, we use three different values:
>
> - $V_c^{\pi}(s)$ means the value function for **uncorrupted policy**.
> - $V_c^{\pi\circ \nu}(s)$ means the value function for **corrupted policy**.
> - $\tilde{V}_c^{\pi,\nu}(s)$
>   is defined as
>
>   $$
>   \begin{align}
>   \tilde{V}_ c^{\pi,\nu}(s) = \mathbb{E}_ {a\sim\pi(\cdot|\nu(s)),s'\sim p(\cdot|s,a)}[r(s,a)+\gamma V_c^{\pi}(s')]
>   \end{align}
>   $$
>
>   It denotes the value when the attacker only corrupts **one state** $s$.
>
> > Q3.3, Q3.4: TV distance.
>
>
> According to the [textbook Sec. 4.1, page 48, Proposition 4.2](http://www.cs.cmu.edu/~15859n/RelatedWork/MarkovChains-MixingTimes.pdf) or [Wikipedia](https://en.wikipedia.org/wiki/Total_variation_distance_of_probability_measures), the TV distance has a coefficient of $1/2$ when the measurement space (in our case, the action space) is countable.
> Therefore, we have the coefficient 2 in Eq. (37) when replacing the L1 norm with TV distance.
>
> (continued)

---

### Public Comment · ~Ezgi_Korkmaz2 · 2023-02-15
**Acknowledgement of Recent Studies**

This paper needs to refer recent studies [1,2,3] on adversarial deep reinforcement learning. More in particular, it has already been shown that certified adversarial training techniques are vulnerable to many different sets of attacks from perturbations that can transfer [2] to natural directions [1]. This study at a minimum should acknowledge and refer to these studies.

[1] Adversarial Robust Deep Reinforcement Learning Requires Redefining Robustness. AAAI Conference on Artificial Intelligence, 2023.

[2] Deep Reinforcement Learning Policies Learn Shared Adversarial Features Across MDPs. AAAI Conference on Artificial Intelligence, 2022.

[3]  Investigating Vulnerabilities of Deep Neural Policies. Conference on Uncertainty in Artificial Intelligence (UAI), Proceedings of Machine Learning Research (PMLR), 2021.

---

### Decision · Program_Chairs · 2023-01-20

**Decision:**

Accept: poster

**Justification For Why Not Higher Score:**

The reviewers generally don't feel very excited about the paper even though they gave scores 6.

The AC finds that the setting studied in the paper seems to be a combination of other more basic settings (adversarial attacks, and safe RL), which are already challenging by themselves, and therefore find the paper somewhat incremental. E.g., it's unclear to the AC why we should study adversarial attacks to safe RL algorithms if the community has struggled with finding good algorithms for safe RL (without adversarial attacks).

**Justification For Why Not Lower Score:**

--- "This work first considers state observation disturbance of safe RL, and proposes the attack strategy to make the attacked policy with high return and high cost, which are called as Stealthiness and Effectiveness of attacking methods in this paper."

---"Numerical experiments are extensive to show the effectiveness of the attacker and the robustness of the proposed adversarial training procedure."

**Metareview: Summary, Strengths And Weaknesses:**

Main strength and weakness in the AC's opinion:

strength:

--- "This work first considers state observation disturbance of safe RL, and proposes the attack strategy to make the attacked policy with high return and high cost, which are called as Stealthiness and Effectiveness of attacking methods in this paper."

---"Numerical experiments are extensive to show the effectiveness of the attacker and the robustness of the proposed adversarial training procedure."

Weakness:

--- The AC finds that the setting studied in the paper seems to be a combination of other more basic settings (adversarial attacks, and safe RL), which are already challenging by themselves, and therefore find the paper somewhat incremental. E.g., it's unclear to the AC why we should study adversarial attacks to safe RL algorithms if the community has struggled with finding good algorithms for safe RL (without adversarial attacks).

**Note From Pc:**

if the above contains the word "oral" or "spotlight" please see: "oral" presentation means -> notable-top-5% and "spotlight" means -> notable-top-25%. As stated in our emails, we are disassociating presentation type from AC recommendations